# Object-Centric Representation Learning via Probabilistic Superpixel Coding

## Abstract

Although Slot Attention (SA) models are widely adopted for object-centric representation learning, they typically assume a shared initialization distribution from which all slots are randomly sampled. This assumption limits their capacity to learn specialized slots that are consistently associated with particular object categories and remain robust to identity-preserving variations in object appearance. To address this limitation, we introduce Probabilistic Superpixel Coding (PSC), an object-centric representation learning method that replaces random slot initialization with a lookup-free identity code-book initialization. Given a set of object proposals, Probabilistic Superpixel Coding factorizes each object representation into two components: (i) an identity token and (ii) a state vector that captures instance-specific variation. We evaluate Probabilistic Superpixel Coding across object identity-stability measures, out-of-distribution grounding, and downstream compositional reasoning tasks. The results demonstrate that Probabilistic Superpixel Coding learns more stable and reusable object representations than slot-based baselines.

## 1 Introduction

A key step in aligning AI systems with human visual perception is equipping them with a human-like notion of objectness Lake et al. (2017). Humans do not perceive the world as a stream of pixels, but rather as a collection of object entities that maintain their identity despite changes in viewpoint, lighting, scale, orientation, and articulation Rock (1973); Kulkarni et al. (2015); Hinton (1979); Behrens et al. (2018). Despite recent progress, several major challenges remain in object-centric representation learning. One such challenge is *identity binding*: a model should associate multiple observations of the same object with more permanent, canonical characteristics of that object, rather than with transient appearance cues alone. Treisman (1999) argued that solving this challenge is a prerequisite for human-like perceptual binding.

Most recent object-centric learning (OCL) methods are based on slot-based approaches Locatello et al. (2020); Kipf et al. (2021); Kori et al. (2023); Jiang et al. (2023); Elsayed et al. (2022); Seitzer et al. (2022), which begun exhibiting impressive results, showing potential to scale to complex visual scenes. Yet, despite this empirical progress, these methods primarily optimize object discovery, reconstruction, or short-horizon temporal propagation, and do not explicitly model a canonical notion of object identity.

Slot Attention represents a scene as an unordered set of object slots. Each slot is initialized from the same distribution and updated with the same refinement network, so the slot index itself does not encode object type or object identity Locatello et al. (2020); Kori et al. (2023; 2024). This design preserves permutation symmetry: permuting the slots should not change the reconstructed scene Locatello et al. (2020); Kori et al. (2024). Video extensions such as SAVi, SAVi++, and depth-aware slot models improve temporal object discovery, but their standard objectives still optimize scene reconstruction rather than persistent identity assignment Kipf et al. (2021); Kori et al. (2023; 2024). Therefore, the model is not explicitly penalized when two slots swap identities across frames, as long as the reconstructed pixels or features remain correct Locatello et al. (2020); Kori et al. (2023; 2024).

This property makes slots effective for object discovery, but it creates ambiguity for identity learning. A slot can represent the orange object in one frame and a different object in another frame, because the

objective rewards accurate decomposition rather than a stable mapping between object identity and slot index Locatello et al. (2020); Kori et al. (2023; 2024). Similarly, a large object can be split across multiple slots if that decomposition improves reconstruction Locatello et al. (2020); Kori et al. (2024). Thus, standard slot-based models learn object-centric scene decompositions, but they do not guarantee identity-grounded representations: the same object identity is not guaranteed to keep the same slot or the same canonical representation across time or scenesKori et al. (2023; 2024); Kipf et al. (2021).

This instability is closely related to the *Binding Problem*, that is, how internal representations are connected to objects in the real-world and their meaning Harnad (1990); Greff et al. (2020). This problem has two closely related components: segregating the object from the surrounding scene and learning a representation that remains tied to that object despite changes in pose, context, and appearance.[1] Recent advances in class-agnostic segmentation and tracking suggest that the segregation component is now handled increasingly well by modern segmentation models.

We therefore focus on the representation side of the Binding Problem. We assume object proposals and ask a different question: how should a model represent an object so that the same object is mapped to the same reusable identity despite frame-specific changes in appearance?.

To address this challenge, we propose Probabilistic Superpixel Coding (PSC), a grounded object-centric representation method that factors each object into two complementary components: a reusable identity code and a frame-specific state vector. The identity component is obtained by aggregating evidence across time and mapping it to a discrete identity token, encouraging the same object to be represented by the same canonical code across frames and viewpoints. The state component captures transient attributes such as pose, motion, and local appearance. In contrast to standard slot-based objectives, which make identity only an implicit by-product of successful grouping or reconstruction, our formulation makes temporal identity consistency an explicit modeling target.

**Contributions.** Our core idea is to replace implicit, exchangeable slot binding with an explicit identity representation that persists across observations. We argue that binding temporary object states to their corresponding permanent object identities can be understood as learning a vocabulary of grounded, canonical object representations. Vector quantization provides a natural mechanism for learning this vocabulary: it maps continuous object representations to a finite set of shared discrete identity codes, encouraging objects with the same underlying identity to reuse the same canonical representation across frames. This reduces the ambiguity of slot permutation and supports more stable cross-frame object binding. To this end, we combine a shared discrete identity vocabulary, cross-frame correspondence, and a factorized identity/state representation to learn object representations that are grounded and temporally stable.

Our main contributions are as follows:

- We introduce a grounded identity-tokenization method for object-centric learning. Instead of relying on the standard slot-based objective to induce stable identity implicitly, the model maps temporally aggregated object evidence to a reusable discrete identity code, yielding a canonical representation that can persist across frames and appearance changes.

- We make temporal identity consistency an explicit modeling objective. The method matches grounded object observations across frames and aggregates identity evidence over time, rather than depending on exchangeable slots to preserve identity implicitly through reconstruction or temporal propagation alone.

- We factorize each object representation into time-invariant identity and time-varying state. This separation allows the method to preserve stable object-level information while independently modelling transient properties such as pose, motion, and local appearance.

---

[1]Neural networks often rely on surface-level statistical regularities rather than underlying concepts, which can hinder systematic generalization.

## 2 Related Work

**Object-Centric Learning.**  Object-centric learning (OCL) represents scenes as compositions of objects. Early methods such as MONET Burgess et al. (2019), IODINE Greff et al. (2019), GENESIS Engelcke et al. (2020), and related compositional generative models Greff et al. (2017); Kosiorek et al. (2018); Lin et al. (2020) learn object-like factors through iterative inference or structured decoding. Slot Attention Locatello et al. (2020) made this paradigm scalable and inspired extensions for images and video Engelcke et al. (2021); Wang et al. (2023); Singh et al. (2022); Elsayed et al. (2022); Singh et al. (2021); Jiang et al. (2023); Seitzer et al. (2022). While effective on controlled benchmarks, these methods often rely on exchangeable slots, which can make object identity unstable across frames, scenes, and viewpoints Greff et al. (2020). Recent work also suggests that, with strong class-agnostic segmenters, object discovery itself may no longer be the main bottleneck Rubinstein et al. (2025); Qi et al. (2022b). Our work is complementary to such proposal or tracking pipelines: rather than solving proposal generation, we study how to convert grounded object observations into temporally persistent object representations.

**Discrete Representation Learning.**  Discrete latent-variable models map continuous features to discrete codes, as in VQ-VAE Van Den Oord et al. (2017), Gumbel-Softmax Jang et al. (2016); Maddison et al. (2016), and later codebook-based extensions Esser et al. (2021); Gu et al. (2022); Ramesh et al. (2021). In PSC, we adopt the lookup-free tokenizer introduced in MAGVIT-v2 Yu et al. (2023), which replaces explicit codebook lookup with binary lookup-free quantization. While MAGVIT-v2 uses this tokenizer for image/video generation and compression, PSC repurposes it for object-centric representation learning: temporally aggregated object evidence is mapped to a reusable discrete identity token, which initializes the identity branch and is paired with a continuous state vector for reconstruction and temporal reasoning.

**Compositional Reasoning and Grounded Representations.**  Compositional reasoning benefits from object representations that are stable and reusable across scenes Greff et al. (2020). Prior OCL methods can decompose scenes into object-like factors Greff et al. (2017); Kosiorek et al. (2018); Crawford & Pineau (2019); Burgess et al. (2019); Greff et al. (2019); Lin et al. (2020); Locatello et al. (2020); Engelcke et al. (2020); Emami et al. (2021); Kipf et al. (2021); Seitzer et al. (2022); Elsayed et al. (2022), but exchangeable slots do not explicitly enforce persistent identity. PSC addresses this limitation by factorizing each grounded object into a discrete identity code and a continuous state variable, with explicit temporal aggregation across frames. This makes PSC particularly suited to settings where the main challenge is not discovering objects from scratch, but maintaining consistent identity for downstream temporal reasoning and compositional transfer Hu et al. (2024); Acuna et al. (2025).

**Segmentation and tracking foundation models.**  Recent foundation models such as SAM 2 Ravi et al. (2024) and SAM 3 Carion et al. (2025) improve promptable video segmentation and concept-based segmentation, respectively. These models are designed to produce masks or tracks, but they do not by themselves enforce a persistent latent identity representation for an object across time. PSC addresses a different problem: given grounded object observations, it learns a factorized representation with a reusable discrete identity code and a time-varying state, so that the same object can maintain a stable representation across frames despite changes in pose, motion, or appearance.

**Superpixel-Guided State Encoding.**  Recent works use superpixels or region structure for sparse spectral–spatial coding Fan et al. (2017), probabilistic color modeling Lin et al. (2018), or cross-modal self-supervision Wang et al. (2025), while related SSL and causal multi-modal methods focus on universal or causally grounded representations Qiang et al. (2024); Wang et al. (2024). In contrast, PSC uses superpixels inside the *state encoder*: ViT patch features are organized through a patch-affinity graph and pooled into object-level state latents. This allows PSC to separate persistent identity from time-varying state, giving it an advantage in dynamic object-centric video settings where pose, motion, appearance, and occlusion change over time.

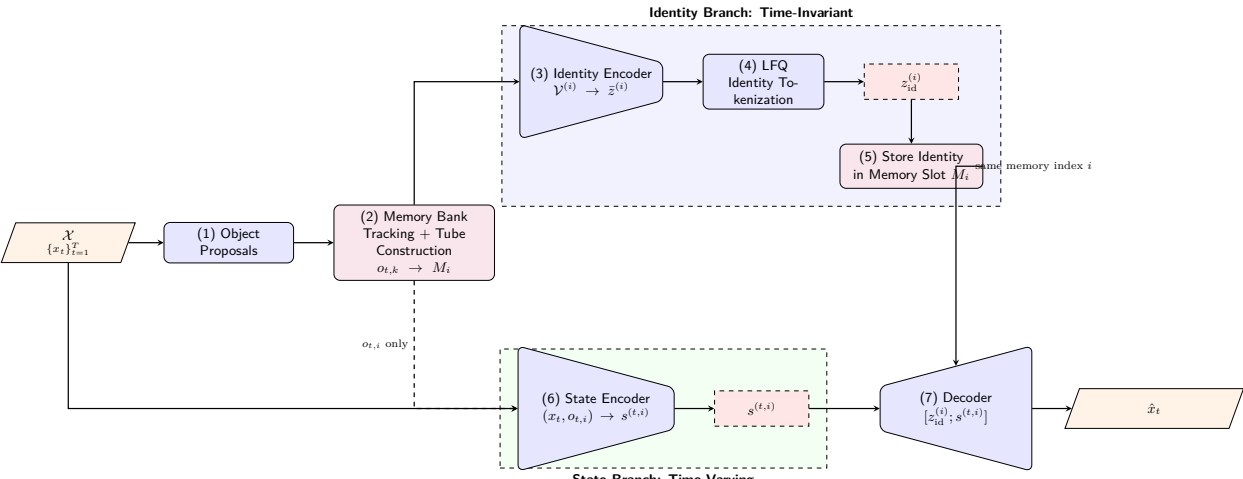

Figure 1: **PSC**: high-level pipeline for disentangled object-centric identity and state representation. The input video is first converted into object proposals. A dynamic memory bank tracks proposals across frames and groups them into persistent object tubes indexed by memory slots $M_i$. The identity branch encodes each tube into a time-invariant representation and maps it through LFQ to a discrete identity code $z_{\text{id}}^{(i)}$, which is stored in the corresponding memory slot. The state branch separately encodes the current frame and the proposal associated with memory slot $i$ into a time-varying state latent $s^{(t,i)}$. The state encoder receives the current proposal $o_{t,i}$ but not the identity code, preventing identity information from being directly injected into the state representation. The decoder reconstructs the frame from the memory-paired representation $[z_{\text{id}}^{(i)}; s^{(t,i)}]$, where pairing is determined by the shared memory index $i$.

## 3 Method

### 3.1 Object Proposals

We propose Probabilistic Superpixel Coding (PSC), an object-centric method for learning stable object identities. Standard Slot Attention represents a scene using exchangeable slots, allowing any slot to bind to any object. While this supports object discovery, it does not ensure that the same slot represents the same object across observations. A reconstruction loss only requires accurate input reconstruction, not consistent identity assignment. As a result, slots may switch between objects or encode short-term cues such as position, color, texture, or pose.

PSC addresses this limitation by explicitly separating object identity from observation-specific state. Given observations $\mathcal{X} = \{x_t\}_{t=1}^T$, each object is factorized into a persistent lookup-free discrete identity token and a continuous state vector. The identity token captures stable object information, while the state vector captures changing attributes such as pose, position, scale, and local appearance. Using vector quantization, aggregated object evidence is mapped to a reusable discrete identity code, creating an identity bottleneck that encourages stable identity representation. The decoder reconstructs the input from the combined identity–state representation, as shown in Fig. 1.

### 3.2 Object Proposals

For each frame $x_t \in [0,1]^{3 \times H \times W}$, where $H$ and $W$ denote height and width, we obtain $K$ object masks $\{m_{t,k}\}_{k=1}^K$ using an external class-agnostic segmentation model such as Qi et al. (2022b); Kirillov et al. (2023); Ravi et al. (2024); Carion et al. (2025). Here, $K$ is the number of object proposals in frame $t$, and $m_{t,k} \in \{0,1\}^{H \times W}$ is the binary mask for object $k$. Each mask is converted into an RGB-A object-

focused input $x_{t,k}$ using AlphaCLIP Sun et al. (2024), where the $\alpha$ channel highlights the target object while preserving scene context. These object proposals provide the inputs to the identity encoder.

To maintain object identity across time, we use a memory bank that associates frame-level object proposals belonging to the same physical object. Specifically, each memory slot $M_i$ stores the temporally grouped proposals of one persistent object identity. The memory bank therefore converts frame-level proposals into object-level video tubes:

$$\mathcal{V}^{(i)} = \{x_{t,k} : x_{t,k} \to M_i\}, \tag{1}$$

where $\mathcal{V}^{(i)}$ denotes the sequence of RGB-A object-focused inputs associated with object identity $i$.

### 3.3 Identity Encoder

The identity encoder produces a *time-invariant* representation for each persistent object. After the memory bank groups frame-level object proposals into object-level video tubes, each tube $\mathcal{V}^{(i)}$ contains the RGB-A object-focused inputs associated with object identity $i$. The identity encoder processes all proposals in this tube and aggregates them into a single stable identity embedding.

For each object-focused input $x_{t,k} \in \mathcal{V}^{(i)}$, we extract a continuous identity feature using the Perception Encoder (PE) Bolya et al. (2025):

$$z_{t,k} = q_{\phi_z}(x_{t,k}) \in \mathbb{R}^D. \tag{2}$$

Here, $z_{t,k}$ captures the appearance and shape evidence of the object proposal at frame $t$. Since individual frames may contain pose changes, partial occlusion, or viewpoint variation, we aggregate all features assigned to the same memory:

$$\bar{z}^{(i)} = \frac{1}{|\mathcal{V}^{(i)}|} \sum_{x_{t,k} \in \mathcal{V}^{(i)}} z_{t,k}. \tag{3}$$

The aggregated feature $\bar{z}^{(i)}$ serves as the continuous identity representation of object $i$. This temporal aggregation makes the identity code stable across frames and robust to late object entry, temporary occlusion, and appearance changes.

We then map the aggregated identity feature to a compact lookup-free discrete identity code similar to Yu et al. (2023):

$$a^{(i)} = g_{\phi_z}\left(\bar{z}^{(i)}\right) \in \mathbb{R}^B, \qquad B = \log_2 K_{\text{id}}. \tag{4}$$

The binary LFQ code is obtained by thresholding each dimension:

$$b_r^{(i)} = 2\,\mathbf{1}_{\{a_r^{(i)} \geq 0\}} - 1, \qquad r = 1, \dots, B. \tag{5}$$

The corresponding integer identity token is:

$$\text{Index}\left(b^{(i)}\right) = \sum_{r=1}^{B} 2^{r-1}\mathbf{1}_{\{b_r^{(i)} > 0\}}. \tag{6}$$

During training, we use a straight-through estimator to pass gradients through the binary quantizer:

$$\tilde{b}^{(i)} = a^{(i)} + \text{sg}\left(b^{(i)} - a^{(i)}\right). \tag{7}$$

Finally, the quantized identity code is projected into the decoder space:

$$z_{\text{id}}^{(i)} = W_{\text{id}}\tilde{b}^{(i)} + c_{\text{id}}. \tag{8}$$

The resulting $z_{\text{id}}^{(i)}$ is shared across all frames of object $i$ and is used as the persistent identity anchor for both state estimation and reconstruction.

### 3.4 State Encoder

The state encoder models the *time-varying* configuration of each object, such as pose, position, scale, deformation, occlusion, and motion-related appearance changes. For each frame $t$, the segmentation model produces a set of object masks

$$\mathcal{M}_t = \{m_{t,k}\}_{k=1}^{K_t}, \qquad m_{t,k} \in [0,1]^{H \times W}.$$

The memory bank assigns each frame-level mask $m_{t,k}$ to a persistent memory slot $M_i$. After assignment, we denote by $m_{t,i}$ the mask associated with memory slot $i$ in frame $t$. The state encoder then produces one frame-specific state vector $s^{(t,i)}$ for each visible memory slot.

Unlike the identity encoder, which aggregates evidence across time, the state encoder is conditioned only on the current frame and the current mask. The mask defines the spatial support of the object but is not treated as an identity representation. Given frame features

$$H_t = F_\psi(x_t), \tag{9}$$

we extract an object-level state feature using mask-conditioned pooling:

$$h_{t,i} = \text{MaskPool}(H_t, m_{t,i}). \tag{10}$$

We also encode mask geometry as

$$g_{t,i} = \gamma(m_{t,i}), \tag{11}$$

where $\gamma(\cdot)$ captures spatial properties such as area, centroid, shape, and extent.

The state posterior is parameterized as

$$q_{\phi_s}\left(s^{(t,i)} \mid x_t, m_{t,i}\right) = \mathcal{N}\left(\mu_{t,i}, \text{diag}(\sigma_{t,i}^2)\right), \tag{12}$$

where

$$(\mu_{t,i}, \log \sigma_{t,i}^2) = E_{\text{st}}(h_{t,i}, g_{t,i}). \tag{13}$$

The state latent is sampled using the reparameterization trick:

$$s^{(t,i)} = \mu_{t,i} + \sigma_{t,i} \odot \epsilon, \qquad \epsilon \sim \mathcal{N}(0, I). \tag{14}$$

Thus, the state encoder produces a separate time-varying state vector for each object mask assigned to a memory slot, while persistent identity is handled separately by the memory-associated LFQ identity code.

### 3.5 Decoder

For each object $k$ in frame $t$, the spatial broadcast decoder Watters et al. (2019); Greff et al. (2019) receives two factorized latent variables: a projected LFQ identity representation $z_{\text{id}}^{(k)}$ and a frame-specific continuous state latent $s^{(t,k)}$.

The LFQ identity code is first projected into the decoder space:

$$z_{\text{id}}^{(k)} = W_{\text{id}}\tilde{b}^{(k)} + c_{\text{id}}, \tag{15}$$

where $\tilde{b}^{(k)}$ is the straight-through version of the lookup-free binary identity code.

The projected LFQ identity representation and the state latent are concatenated and passed to a shared decoder:

$$u^{(t,k)} = \left[z_{\text{id}}^{(k)}; s^{(t,k)}\right]. \tag{16}$$

The shared decoder reconstructs the object appearance from the joint identity–state representation:

$$(\hat{v}^{(t,k)}, \hat{\alpha}^{(t,k)}) = D_\theta\left(u^{(t,k)}\right), \tag{17}$$

where $\hat{v}^{(t,k)}$ is the RGB reconstruction of object $k$ in frame $t$, and $\hat{\alpha}^{(t,k)}$ is the corresponding alpha mask.

The object reconstructions are composed into the final frame by normalizing the alpha masks across objects:

$$\hat{x}_t = \sum_k \mathrm{softmax}_k \left( \hat{\alpha}^{(t,k)} \right) \hat{v}^{(t,k)}. \tag{18}$$

Reconstruction is conditioned jointly on two latent factors. In our case, $z_{\mathrm{id}}^{(k)}$ represents the projected time-invariant LFQ identity code, while $s^{(t,k)}$ represents the time-varying object state, including pose, position, motion, deformation, occlusion, and local appearance. The two variables are learned by separate encoders and are only combined at decoding time.

### 3.6 Objective

We optimize an ELBO-style objective with lookup-free quantization for identity and a contrastive separation term between identity and state. The temporal identity-consistency loss is removed. Since each memory slot stores a persistent identity code and the decoder pairs this identity code with the state code from the same memory slot, temporal identity consistency is enforced architecturally through the memory bank.

The full training objective is

$$
\begin{aligned}
\mathcal{L}_{\mathrm{PSC\text{-}LFQ}} = & -\mathbb{E}_{\mathbf{s} \sim q_{\phi_s}(\mathbf{s}|\mathbf{x},\mathbf{o})} \Big[ \log p_\theta \big( \mathbf{x} \mid \mathbf{s}, \mathbf{z}_{\mathrm{id}}, \mathbf{o} \big) \Big] \\
& + \lambda_s D_{\mathrm{KL}} \big( q_{\phi_s}(\mathbf{s} \mid \mathbf{x}, \mathbf{o}) \,\|\, p(\mathbf{s}) \big) \\
& + \beta \mathcal{L}_{\mathrm{commit}} + \lambda_{\mathrm{entropy}} \mathcal{L}_{\mathrm{entropy}} \\
& + \lambda_{\mathrm{con}} \mathcal{L}_{\mathrm{con}}^{s,\mathrm{id}}.
\end{aligned}
\tag{19}
$$

Here, $\mathbf{x}$ denotes the input video, $\mathbf{o}$ denotes the set of object proposals, $\mathbf{s}$ denotes the continuous time-varying state latents, and $\mathbf{z}_{\mathrm{id}}$ denotes the LFQ identity representations stored in the memory bank. The state posterior is now conditioned on the frame and object proposal:

$$q_{\phi_s}(\mathbf{s} \mid \mathbf{x}, \mathbf{o}), \tag{20}$$

rather than on identity codes. The decoder likelihood reconstructs the video from memory-paired identity and state representations:

$$p_\theta(\mathbf{x} \mid \mathbf{s}, \mathbf{z}_{\mathrm{id}}, \mathbf{o}). \tag{21}$$

The identity encoder maps temporally aggregated object evidence to LFQ logits $\mathbf{a}^{(i)} \in \mathbb{R}^B$ for memory slot $i$, where $B = \log_2 K_{\mathrm{id}}$ and $K_{\mathrm{id}}$ is the size of the identity vocabulary. The LFQ quantizer produces a binary identity code:

$$\mathbf{b}^{(i)} = q_{\mathrm{LFQ}}(\mathbf{a}^{(i)}), \qquad b_r^{(i)} = 2\mathbf{1}\{a_r^{(i)} \geq 0\} - 1, \quad r = 1, \ldots, B. \tag{22}$$

During training, gradients are passed through the non-differentiable binary quantizer using a straight-through estimator:

$$\widetilde{\mathbf{b}}^{(i)} = \mathbf{a}^{(i)} + \mathrm{sg}\big(\mathbf{b}^{(i)} - \mathbf{a}^{(i)}\big), \tag{23}$$

where $\mathrm{sg}(\cdot)$ denotes the stop-gradient operation. The projected identity representation stored in the memory bank and used by the decoder is

$$\mathbf{z}_{\mathrm{id}}^{(i)} = W_{\mathrm{id}}\widetilde{\mathbf{b}}^{(i)} + \mathbf{c}_{\mathrm{id}}. \tag{24}$$

The LFQ commitment loss encourages the continuous logits to commit to their assigned binary identity codes:

$$\mathcal{L}_{\mathrm{commit}} = \frac{1}{N_{\mathrm{mem}}} \sum_{i=1}^{N_{\mathrm{mem}}} \left\| \mathbf{a}^{(i)} - \mathrm{sg}\big[\mathbf{b}^{(i)}\big] \right\|_2^2, \tag{25}$$

where $N_{\mathrm{mem}}$ is the number of active memory slots in the minibatch.

We also use the LFQ entropy codebook-utilization loss:

$$\mathcal{L}_{\text{entropy}} = \mathbb{E}\big[H(q_{\text{LFQ}}(\mathbf{a}))\big] - H\big(\mathbb{E}[q_{\text{LFQ}}(\mathbf{a})]\big). \tag{26}$$

In the binary LFQ case, this can be written using soft probabilities $p_r^{(i)} = \sigma(a_r^{(i)}/\tau)$:

$$\mathcal{L}_{\text{entropy}} = \frac{1}{N_{\text{mem}}} \sum_{i=1}^{N_{\text{mem}}} \sum_{r=1}^{B} h(p_r^{(i)}) - \sum_{r=1}^{B} h(\bar{p}_r), \qquad \bar{p}_r = \frac{1}{N_{\text{mem}}} \sum_{i=1}^{N_{\text{mem}}} p_r^{(i)}, \tag{27}$$

where $h(p) = -p \log p - (1-p) \log(1-p)$.

Finally, we add a contrastive identity–state separation loss to discourage the state latent from encoding identity-specific information. Since the state encoder no longer receives $z_{\text{id}}^{(i)}$, this loss acts as an additional disentanglement constraint rather than as an assignment mechanism.

We first project state and identity into a shared contrastive space:

$$\mathbf{u}_{t,i} = \frac{r_s(\mathbf{s}^{(t,i)})}{\|r_s(\mathbf{s}^{(t,i)})\|_2}, \qquad \mathbf{v}_i = \frac{r_{\text{id}}(\mathbf{z}_{\text{id}}^{(i)})}{\|r_{\text{id}}(\mathbf{z}_{\text{id}}^{(i)})\|_2}. \tag{28}$$

The separation loss is

$$\mathcal{L}_{\text{con}}^{s,\text{id}} = \frac{1}{\sum_i |\mathcal{T}_i|} \sum_i \sum_{t \in \mathcal{T}_i} \big[\max\big(0, \mathbf{u}_{t,i}^{\top} \text{sg}[\mathbf{v}_i] - m\big)\big]^2, \tag{29}$$

where $\mathcal{T}_i$ is the set of frames where memory slot $i$ is visible and $m$ is a similarity margin. We apply stop-gradient to the identity projection in this term so that the contrastive loss primarily removes identity information from the state representation, while the identity code itself remains governed by LFQ, commitment, entropy utilization, and reconstruction.

The scalars $\lambda_s$, $\beta$, $\lambda_{\text{entropy}}$, and $\lambda_{\text{con}}$ control the state KL regularization, LFQ commitment loss, entropy utilization loss, and identity–state contrastive separation loss, respectively. We minimize Eq. equation 19 with respect to the state encoder, identity encoder, decoder, LFQ projection parameters, and contrastive projection heads.

## 4 Experiments

This section evaluates PSC across four experiments: object identity stability, identity–state factorization, OOD grounding and sample-efficient transfer, and visual reasoning transfer. Together, these experiments test whether the model maintains stable object identity over time, separates identity from time-varying state, generalizes under appearance and context shift, and transfers to downstream reasoning tasks with limited supervision.

We also report PSC variants to isolate key components: PSC-noLFQ removes lookup-free quantization, PSC-noTempAgg removes temporal aggregation, and PSC-noSPState removes the superpixel-guided state encoder. Additional implementation details and non-regression results are provided in the appendix.

### 4.1 Object Identity Stability

OCL methods are commonly evaluated in terms of unsupervised object discovery, where performance is measured by the ability to segment individual object instances in a scene. We explore whether object-centric models can bind representation to specific object types. We define stable object identity operationally as *persistent latent-object correspondence*: the same object should remain assigned to the same slot and should maintain a stable representation. By stable representation, we mean that the latent features assigned to the same object remain consistent, exhibiting low intra-object variance or standard deviation.

Table 1: Category-level representation stability on MS COCO. Object instances are grouped by semantic category. RD measures intra-category representation deviation; lower is better. IDR@1 measures whether the nearest retrieved representation belongs to the same object category; higher is better.

| METHOD | RD ↓ | IDR@1 ↑ |
|---|---|---|
| Slot Attention Locatello et al. (2020) | 0.312±0.021 | 54.8±2.6 |
| SAVI Kipf et al. (2021) | 0.274±0.019 | 59.6±2.3 |
| SAVI++ Elsayed et al. (2022) | 0.251±0.017 | 62.7±2.1 |
| SOLD Mosbach et al. (2024) | 0.238±0.016 | 64.2±2.0 |
| CoSA-GSD Kori et al. (2023) | 0.229±0.015 | 65.5±1.9 |
| OCCAM Rubinstein et al. (2025) | 0.217±0.014 | 67.1±1.8 |
| PSC-noTempAgg | 0.224±0.015 | 66.4±1.9 |
| PSC-noCodebook | 0.204±0.013 | 69.3±1.7 |
| PSC | **0.126±0.010** | **85.8±1.4** |

**Datasets.** We utilized the MS COCO Lin et al. (2014) dataset for its diverse collection of real-world images, each featuring multiple co-occurring objects. This dataset poses a significant challenge for object-centric learning models due to the complexity of the scenes. Furthermore, we used the synthetic datasets MOVi-C and MOVi-E Greff et al. (2022), which contain approximately 1000 realistic 3D-scanned objects. MOVi-C includes scenes with 3-10 objects, whereas MOVi-E contains scenes with 11-23 objects per scene.

**Setup.** For MS COCO, we evaluated the stability of category-level representations. Since COCO contains static images and does not provide temporal object tracks, we group object instances by their semantic label. For each image, we match each ground-truth instance mask to the predicted slot with the highest mask IoU and use the corresponding slot embedding as the object representation. Images with multiple objects are handled at the instance level and multiple instances of the same category are eliminated.

For MOVi-C and MOVi-E, we evaluate the stability of the temporal identity. These datasets provide frame-level masks, visibility information, and persistent object IDs. For each visible object in each frame, we assign the object to the predicted slot with the highest mask IoU and track both the slot assignment and the slot representation across time.

**Metrics.** We report three identity-stability metrics. **Representation Deviation** (RD) measures the temporal variation of the matched object representation:

$$\text{RD}_i = \frac{1}{|\mathcal{T}_i|} \sum_{t \in \mathcal{T}_i} \|z_i^t - \bar{z}_i\|_2, \quad \bar{z}_i = \frac{1}{|\mathcal{T}_i|} \sum_{t \in \mathcal{T}_i} z_i^t.$$

**Identification Rate** (IDR@1) measures whether an object representation at frame $t$ retrieves the same object at a later frame $t + \Delta$:

$$\text{IDR@1} = \frac{1}{|\mathcal{Q}|} \sum_{(i,t,\Delta) \in \mathcal{Q}} \mathbb{1} \left[ \arg\max_j \cos(z_i^t, z_j^{t+\Delta}) = i \right].$$

**Slot Assignment Consistency** (SAC) measures how often object $O_i$ is assigned to its dominant slot:

$$\text{SAC}_i = \frac{1}{|\mathcal{T}_i|} \sum_{t \in \mathcal{T}_i} \mathbb{1}[s_t(i) = s_i^\star], \quad s_i^\star = \text{mode}(\{s_t(i)\}_{t \in \mathcal{T}_i}).$$

## 4.2 Identity–State Factorization

This experiment evaluates whether PSC separates persistent object identity from time-varying object state. Following our operational definition, object identity is not treated as semantic category recognition. Instead, the identity token $z_{\text{id}}$ should remain stable for the same physical object throughout the video, while the state representation $s_t$ should capture frame-dependent variation such as position, motion, orientation, visibility, and appearance changes.

Table 2: Object identity stability on MOVi-C/D/E. Identity is evaluated as persistent latent-object correspondence within each video. SAC and IDR@1 are reported as percentages. RD measures latent representation deviation; lower is better.

| METHOD | SAC ↑ | RD ↓ | IDR@1 ↑ |
|---|---|---|---|
| Slot Attention Locatello et al. (2020) | 61.4±2.3 | 0.284±0.018 | 58.7±2.5 |
| SAVI Kipf et al. (2021) | 69.8±1.9 | 0.231±0.015 | 66.2±2.1 |
| SAVI++Elsayed et al. (2022) | 73.6±1.7 | 0.207±0.014 | 70.4±1.8 |
| SOLD Mosbach et al. (2024) | 75.1±1.6 | 0.194±0.012 | 72.8±1.7 |
| CoSA-GSD Kori et al. (2023) | 76.4±1.5 | 0.187±0.011 | 73.9±1.6 |
| OCCAM Rubinstein et al. (2025) | 78.1±1.4 | 0.171±0.010 | 75.8±1.5 |
| PSC-noTempAgg | 77.3±1.5 | 0.176±0.011 | 74.9±1.6 |
| PSC-noCodebook | 79.2±1.4 | 0.163±0.010 | 76.5±1.5 |
| PSC | **86.5±1.1** | **0.118±0.008** | **83.7±1.2** |

Table 3: Identity–state factorization on MOVi-A/B. Identity is evaluated as stability of $z_{\mathrm{id}}$ for the same object across the video. State is evaluated as the ability of $s_t$ to capture frame-level variation. IDev is lower when identity tokens remain stable. SVar is higher when state representations vary with the object.

| METHOD | IDEV ↓ | SVAR ↑ |
|---|---|---|
| CoSA/GSD (Kori et al., 2023) | 0.238±0.014 | 0.421±0.018 |
| PSC-noLFQ | 0.184±0.011 | 0.447±0.016 |
| PSC-noTempAgg | 0.207±0.012 | 0.439±0.017 |
| PSC | **0.091±0.007** | **0.486±0.014** |

**Setup.** We evaluate on MOVi-A/B, which provide controlled object attributes and frame-level object information. For each object $O_i$, we collect its visible trajectory $\mathcal{T}_i$. At each visible frame $t$, the model produces an identity token $z_{\mathrm{id}}^t(i)$ and a state representation $s_t(i)$.

A successful factorization should satisfy two properties: (i) $z_{\mathrm{id}}^t(i)$ should be constant across $t \in \mathcal{T}_i$, and (ii) $s_t(i)$ should vary with the object's changing frame-level state.

**Metrics.** We report four metrics. **Identity Deviation** measures how much the identity token changes for the same object:

$$\mathrm{IDev}_i = \frac{1}{|\mathcal{T}_i|} \sum_{t \in \mathcal{T}_i} \left\| z_{\mathrm{id}}^t(i) - \bar{z}_{\mathrm{id}}(i) \right\|_2, \quad \bar{z}_{\mathrm{id}}(i) = \frac{1}{|\mathcal{T}_i|} \sum_{t \in \mathcal{T}_i} z_{\mathrm{id}}^t(i).$$

Lower values indicate a more stable identity token.

**State Variation** measures whether the state branch changes over time:

$$\mathrm{SVar}_i = \frac{1}{|\mathcal{T}_i|} \sum_{t \in \mathcal{T}_i} \left\| s_t(i) - \bar{s}(i) \right\|_2.$$

Higher values indicate that the state branch captures temporal variation.

### 4.3 Out-of-distribution Grounding

This experiment tests whether object identity remains stable under out-of-distribution shifts. Identity is evaluated as physical object persistence, not semantic recognition: the same object should keep a stable identity token across time despite changes in appearance, background, lighting, clutter, and occlusion.

Table 4: OOD grounding and sample-efficient transfer from MOVi/Kubric to OVIS. OOD FG-ARI and OOD mIoU measure object grounding. OOD IDR@1 and OOD IDev measure identity persistence across visible annotated frames. Gap is the source-to-target FG-ARI drop. Values are placeholders and should be replaced with measured mean ± std. over seeds.

| METHOD | OOD FG-ARI ↑ | OOD mIoU ↑ | OOD IDR@1 ↑ | OOD IDev ↓ | GAP ↓ | 10-shot ↑ | Few-shot AUC ↑ |
|---|---|---|---|---|---|---|---|
| Slot Attention (Locatello et al., 2020) | 54.2±2.5 | 42.6±2.3 | 35.4±2.8 | 0.312±0.020 | 21.3±2.4 | 50.2±2.6 | 56.9±2.3 |
| DINOSAUR (Seitzer et al., 2022) | 62.7±2.1 | 50.4±2.0 | 44.8±2.4 | 0.276±0.018 | 17.4±2.1 | 60.8±2.2 | 66.3±2.0 |
| VideoSAUR (Zadaianchuk et al., 2023) | 69.8±1.8 | 58.7±1.7 | 56.9±2.1 | 0.224±0.015 | 14.2±1.8 | 68.7±1.9 | 73.5±1.7 |
| CoSA/GSD (Kori et al., 2023) | 72.4±1.7 | 61.3±1.6 | 60.2±2.0 | 0.207±0.014 | 12.9±1.7 | 72.5±1.8 | 76.4±1.6 |
| PSC-noLFQ | 74.1±1.5 | 63.6±1.5 | 64.8±1.8 | 0.181±0.012 | 12.4±1.6 | 74.6±1.7 | 78.8±1.5 |
| PSC-noTempAgg | 75.6±1.4 | 64.9±1.4 | 61.5±1.9 | 0.203±0.013 | 11.7±1.5 | 75.8±1.6 | 79.6±1.4 |
| PSC | **84.2±1.1** | **76.8±1.2** | **78.4±1.3** | **0.104±0.008** | **7.4±1.1** | **83.7±1.2** | **87.1±1.1** |

**Setup.** We evaluate zero-shot on OVIS Qi et al. (2022a), a real-world occluded video instance segmentation benchmark with temporally consistent instance masks. OVIS contains real videos, camera motion, clutter, occlusion, and object disappearance/reappearance.

We evaluate only frames with annotated visible masks. For each object $O_i$, its visible trajectory is

$$\mathcal{T}_i = \{t \mid M_i^t \text{ is annotated and } |M_i^t| > A_{\min}\}.$$

Predicted objects are matched to ground-truth masks using Hungarian matching with mask IoU. A match is accepted when IoU exceeds $\tau_{\mathrm{IoU}}$. Ground-truth masks are used only for evaluation, not as model input. All baselines use the same resolution, frame sampling, number of slots/proposals, and external mask pipeline where applicable.

**Metrics.** We report OOD FG-ARI and OOD mIoU to measure object grouping and mask grounding. To evaluate identity persistence, we report OOD IDR@1, where an identity token from frame $t$ must retrieve the same object at a later visible frame $t'$. We also report identity-token deviation:

$$\mathrm{IDev}_i = \frac{1}{|\mathcal{T}_i|} \sum_{t \in \mathcal{T}_i} \left\| z_{\mathrm{id}}^t(i) - \bar{z}_{\mathrm{id}}(i) \right\|_2, \quad \bar{z}_{\mathrm{id}}(i) = \frac{1}{|\mathcal{T}_i|} \sum_{t \in \mathcal{T}_i} z_{\mathrm{id}}^t(i).$$

Lower IDev means the same object keeps a more stable identity representation.

For sample-efficient transfer, we freeze the learned representation and train lightweight target heads using $k \in \{10, 50, 100\}$ labelled OVIS tracks. We report 10-shot accuracy and few-shot AUC.

## 4.4 Visual Reasoning and Transfer

We evaluate downstream reasoning transfer on FMNIST2 addition and subtraction tasks. The benchmark tests whether object representations learned on one arithmetic rule transfer to a related rule with limited target supervision. We train each model on either FMNIST2-Add or FMNIST2-Sub and evaluate both in-domain performance and few-shot cross-task transfer with $k$=100 labelled target examples. Accuracy measures task performance, while HMC and rationale F1 evaluate whether the predicted rationale aligns with the relevant objects.

Table 5 shows that high source accuracy alone does not guarantee transfer. CNN, SA, and Block-Slot achieve strong in-domain accuracy but fail under Add→Sub and Sub→Add transfer. Object-centric methods transfer substantially better, indicating that object-level representations are more reusable across rule changes. PSC obtains the best source accuracy, target accuracy, and rationale alignment in both transfer directions, improving Add→Sub target accuracy from 60.24 to 70.13 and Sub→Add target accuracy from 63.29 to 69.13. These results suggest that the learned object representation supports both rule transfer and more faithful object-level rationales.

Table 5: Reasoning transfer on FMNIST2 addition and subtraction. Source columns report in-domain accuracy and rationale alignment. Target columns report few-shot transfer with $k$=100 labelled examples. Lower HMC indicates better rationale alignment.

| METHOD | ADD$_{source}$ | | SUB$_{target}$ | | SUB$_{source}$ | | ADD$_{target}$ | |
|---|---|---|---|---|---|---|---|---|
| | Acc ↑ | HMC ↓ | Acc ↑ | F1 ↑ | Acc ↑ | HMC ↓ | Acc ↑ | F1 ↑ |
| CNN | 97.62 | – | 10.35 | 10.05 | 98.16 | – | 12.35 | 9.50 |
| SA | 97.33 | 0.14 | 11.06 | 9.40 | 97.41 | 0.13 | 8.28 | 7.83 |
| Block-Slot | 98.11 | 0.12 | 9.71 | 9.10 | 97.42 | 0.14 | 9.61 | 8.36 |
| CoSA | 98.12 | 0.10 | 60.24 | 50.16 | 98.64 | 0.12 | 63.29 | 58.29 |
| AdaSlot | 98.12 | 0.10 | 60.24 | 50.16 | 98.64 | 0.12 | 63.29 | 58.29 |
| SPOT | 98.12 | 0.10 | 60.24 | 50.16 | 98.64 | 0.12 | 63.29 | 58.29 |
| PSC | **99.12** | **0.06** | **70.13** | **58.00** | **99.30** | **0.09** | **69.13** | **69.20** |

## 5 Discussion, Limitations, Conclusion, and Future Work

**Discussion.** Our results suggest that PSC provides a useful mechanism for making object identity more persistent than in exchangeable-slot models. By combining grounded object proposals, temporal aggregation, and LFQ identity tokens, the model separates relatively stable object identity from time-varying state factors such as pose, position, visibility, and local appearance. This distinction is important because accurate reconstruction or segmentation does not necessarily imply stable identity assignment over time. The factorization also suggests a broader form of compositional reuse: while identity codes are object-specific, some state factors, such as "tilted left", "partially occluded", or "moving upward", are not object-specific and could be reused across different object identities.

**Limitations.** Several limitations remain. First, PSC depends on external object proposals and an object-count prior; missed, merged, or fragmented masks can propagate errors into both identity and state representations. Second, although our evaluation includes identity-oriented video metrics, the current experiments are still limited in the length and complexity of the videos considered. Stronger validation on longer, high-resolution, and unconstrained real-world videos is needed to better support claims about long-horizon identity persistence. Third, the current version does not include sufficient qualitative analysis of successes and failures. Important failure cases to visualize include identity swaps between similar objects, identity drift after occlusion, incorrect assignments caused by mask errors, and possible leakage of identity information into the state representation. Finally, the state representation is not explicitly constrained to be object-agnostic, so reusable state concepts may still be entangled with object identity.

**Future Work.** Future work should first evaluate PSC on longer and more complex real-world video datasets with camera motion, clutter, occlusion, object disappearance and re-entry, and visually similar instances. Second, future versions should include qualitative visualizations of masks, memory slots, identity tokens, reconstructions, and identity trajectories to better diagnose where the method succeeds or fails. Third, the state branch could be extended into a reusable vocabulary of object-agnostic state primitives, allowing factors such as orientation, motion, and visibility to transfer across object categories. Finally, an important direction is to reduce dependence on external masks by developing a self-supervised or proposal-light version that learns object discovery, temporal correspondence, and identity–state separation directly from raw video.

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

# A  Derivation of Variational Lower Bound

In this appendix, we derive the variational lower bound used in Eq. equation 19. The model represents each input video $\boldsymbol{x}^{(n)}$ using a continuous state latent $\boldsymbol{s}$ and a lookup-free identity representation $\boldsymbol{z}_{\mathrm{id}}$. The variational lower bound provides the reconstruction and state-regularization terms, while the lookup-free quantization, temporal identity-consistency, and contrastive identity–state separation terms are added as regularizers for stable and disentangled object representations.

## A.1  Variational Lower Bound

For each sample $\boldsymbol{x}^{(n)}$, the identity encoder produces a deterministic identity representation $\boldsymbol{z}_{\mathrm{id}}^{(n)}$. Conditioned on this identity representation, the marginal likelihood is

$$\log p_\theta(\boldsymbol{x}^{(n)} \mid \boldsymbol{z}_{\mathrm{id}}^{(n)}) = \log \int p_\theta(\boldsymbol{x}^{(n)}, \boldsymbol{s} \mid \boldsymbol{z}_{\mathrm{id}}^{(n)})\, d\boldsymbol{s} \tag{30}$$

$$= \log \int q_{\phi_s}(\boldsymbol{s} \mid \boldsymbol{x}^{(n)}) \frac{p_\theta(\boldsymbol{x}^{(n)}, \boldsymbol{s} \mid \boldsymbol{z}_{\mathrm{id}}^{(n)})}{q_{\phi_s}(\boldsymbol{s} \mid \boldsymbol{x}^{(n)})} d\boldsymbol{s} \tag{31}$$

$$\geq \int q_{\phi_s}(\boldsymbol{s} \mid \boldsymbol{x}^{(n)}) \log \frac{p_\theta(\boldsymbol{x}^{(n)}, \boldsymbol{s} \mid \boldsymbol{z}_{\mathrm{id}}^{(n)})}{q_{\phi_s}(\boldsymbol{s} \mid \boldsymbol{x}^{(n)})} d\boldsymbol{s} \tag{32}$$

$$= \int q_{\phi_s}(\boldsymbol{s} \mid \boldsymbol{x}^{(n)}) \log \frac{p_\theta(\boldsymbol{x}^{(n)} \mid \boldsymbol{s}, \boldsymbol{z}_{\mathrm{id}}^{(n)}) p(\boldsymbol{s})}{q_{\phi_s}(\boldsymbol{s} \mid \boldsymbol{x}^{(n)})} d\boldsymbol{s} \tag{33}$$

$$= \mathbb{E}_{\boldsymbol{s} \sim q_{\phi_s}(\boldsymbol{s} \mid \boldsymbol{x}^{(n)})} \left[ \log p_\theta\left(\boldsymbol{x}^{(n)} \mid \boldsymbol{s}, \boldsymbol{z}_{\mathrm{id}}^{(n)}\right) \right] - D_{\mathrm{KL}}\left( q_{\phi_s}(\boldsymbol{s} \mid \boldsymbol{x}^{(n)}) \,\|\, p(\boldsymbol{s}) \right). \tag{34}$$

The inequality follows from Jensen's inequality. Therefore, maximizing the lower bound is equivalent to minimizing the negative reconstruction likelihood and the state KL regularization term.

## A.2  Lookup-Free Identity Representation

For object track $i$, the identity encoder maps the temporally aggregated object feature $\bar{\boldsymbol{z}}^{(i)}$ to LFQ logits:

$$^{(i)} = g_{\phi_z}(\bar{\boldsymbol{z}}^{(i)}) \in \mathbb{R}^B, \qquad B = \log_2 K_{\mathrm{id}}, \tag{35}$$

where $K_{\mathrm{id}}$ is the size of the identity vocabulary. The binary identity code is obtained by independent sign quantization:

$$\boldsymbol{b}^{(i)} = q_{\mathrm{LFQ}}(^{(i)}), \qquad b_r^{(i)} = 2\mathbf{1}[a_r^{(i)} \geq 0] - 1, \quad r = 1, \ldots, B. \tag{36}$$

Since the sign operation is non-differentiable, we use the straight-through estimator

$$\tilde{\boldsymbol{b}}^{(i)} = {}^{(i)} + \mathrm{sg}\left[ \boldsymbol{b}^{(i)} - {}^{(i)} \right], \tag{37}$$

where $\mathrm{sg}[\cdot]$ denotes the stop-gradient operation. The projected identity representation used by the decoder is

$$\boldsymbol{z}_{\mathrm{id}}^{(i)} = W_{\mathrm{id}} \tilde{\boldsymbol{b}}^{(i)} + \boldsymbol{c}_{\mathrm{id}}. \tag{38}$$

## A.3  LFQ Commitment Loss

The LFQ commitment loss encourages the continuous logits $^{(i)}$ to remain close to their assigned binary identity code. Since LFQ does not use a learned embedding codebook, the commitment penalty is applied directly between the pre-quantized logits and the stop-gradient binary code:

$$\mathcal{L}_{\mathrm{commit}} = \frac{1}{N_{\mathrm{trk}}} \sum_{i=1}^{N_{\mathrm{trk}}} \left\| ^{(i)} - \mathrm{sg}\left[ \boldsymbol{b}^{(i)} \right] \right\|_2^2. \tag{39}$$

This term stabilizes LFQ training by preventing the encoder outputs from drifting away from their discrete binary assignments.

## A.4 Entropy Codebook-Utilization Loss

To encourage confident and balanced binary assignments, we use the LFQ entropy codebook-utilization loss:

$$\mathcal{L}_{\text{entropy}} = \mathbb{E}\left[H(q_{\text{LFQ}}())\right] - H\left(\mathbb{E}\left[q_{\text{LFQ}}()\right]\right). \tag{40}$$

The first term penalizes high entropy for individual assignments, encouraging each object to make a confident binary decision. The second term encourages the average assignment distribution to have high entropy, which promotes balanced usage of the identity code space.

For binary LFQ, we approximate the assignment probability of dimension $r$ for track $i$ as

$$p_r^{(i)} = \sigma(a_r^{(i)}/\tau), \qquad \bar{p}_r = \frac{1}{N_{\text{trk}}} \sum_{i=1}^{N_{\text{trk}}} p_r^{(i)}, \tag{41}$$

where $\sigma(\cdot)$ is the sigmoid function and $\tau$ is a temperature parameter. The entropy codebook-utilization loss becomes

$$\mathcal{L}_{\text{entropy}} = \frac{1}{N_{\text{trk}}} \sum_{i=1}^{N_{\text{trk}}} \sum_{r=1}^{B} h(p_r^{(i)}) - \sum_{r=1}^{B} h(\bar{p}_r), \tag{42}$$

where

$$h(p) = -p\log p - (1-p)\log(1-p) \tag{43}$$

is the binary entropy function.

## A.5 Temporal Identity-Consistency Loss

Temporal identity consistency encourages all frame-level proposals assigned to the same object track to agree with the track-level identity representation. Let $\mathcal{M}_i$ denote the set of frame-level proposals assigned to object track $i$, and let $_{t,k}$ denote the LFQ logits predicted from proposal $(t, k)$. The aggregated track-level logits $^{(i)}$ serve as the identity anchor for all proposals in the same track. We define

$$\mathcal{L}_{\text{temp}} = \frac{1}{\sum_i |\mathcal{M}_i|} \sum_i \sum_{(t,k)\in\mathcal{M}_i} \left\| \tanh(_{t,k}) - \text{sg}\left[\tanh(^{(i)})\right] \right\|_2^2. \tag{44}$$

The hyperbolic tangent maps the logits to a soft binary range before comparison. The stop-gradient operation treats the aggregated identity as a fixed temporal anchor. This term discourages identity drift by requiring frame-level identity evidence for the same physical object to remain consistent across time.

## A.6 Contrastive Identity–State Separation Loss

The variational lower bound encourages accurate reconstruction from both $\boldsymbol{s}$ and $\boldsymbol{z}_{\text{id}}$, but reconstruction alone does not guarantee that the two variables encode different information. In particular, the state latent may still encode identity-specific information. To discourage this leakage, we add a contrastive identity–state separation loss.

Because the state latent and identity representation may have different dimensions, we first map them to a shared contrastive space using projection heads:

$$r_s : \mathcal{S} \to \mathbb{R}^{d_c}, \qquad r_{\text{id}} : \mathcal{Z}_{\text{id}} \to \mathbb{R}^{d_c}. \tag{45}$$

For object track $i$ at frame $t$, define the normalized projected embeddings

$$\boldsymbol{u}_{t,i} = \frac{r_s(\boldsymbol{s}^{(t,i)})}{\|r_s(\boldsymbol{s}^{(t,i)})\|_2}, \qquad \boldsymbol{b}_i = \frac{r_{\text{id}}(\boldsymbol{z}_{\text{id}}^{(i)})}{\|r_{\text{id}}(\boldsymbol{z}_{\text{id}}^{(i)})\|_2}. \tag{46}$$

Their inner product

$$\boldsymbol{u}_{t,i}^\top \boldsymbol{b}_i \tag{47}$$

is the cosine similarity between the projected state and identity representations. We then define

$$\mathcal{L}_{\text{con}}^{s,\text{id}} = \frac{1}{\sum_i |\mathcal{T}_i|} \sum_i \sum_{t \in \mathcal{T}_i} \left[ \max \left( 0, \boldsymbol{u}_{t,i}^\top \boldsymbol{b}_i - m \right) \right]^2, \tag{48}$$

where $\mathcal{T}_i$ is the set of visible frames for object track $i$, and $m$ is a similarity margin. When $m = 0$, the loss penalizes positive cosine similarity and encourages approximate orthogonality between state and identity. When $m < 0$, the loss enforces a stronger separation by encouraging the two representations to be anti-correlated.

This term treats the state and identity representations from the same object as factors that should be separated rather than aligned. Therefore, unlike standard contrastive objectives that pull positive pairs together, this loss explicitly pushes the state representation away from the identity representation.

## A.7 Final Training Objective

Taking the negative of the lower bound and adding the LFQ identity losses, temporal consistency loss, and contrastive identity–state separation loss gives the final minimization objective:

$$\begin{aligned}
\mathcal{L}_{\text{PSC-LFQ}} = & -\mathbb{E}_{\boldsymbol{s} \sim q_{\phi_s}(\boldsymbol{s}|\boldsymbol{x})} \left[ \log p_\theta \left( \boldsymbol{x} \mid \boldsymbol{s}, \boldsymbol{z}_{\text{id}} \right) \right] \\
& + \lambda_s D_{\text{KL}} \left( q_{\phi_s} \left( \boldsymbol{s} \mid \boldsymbol{x} \right) \| p(\boldsymbol{s}) \right) \\
& + \beta \mathcal{L}_{\text{commit}} + \lambda_{\text{entropy}} \mathcal{L}_{\text{entropy}} + \lambda_{\text{temp}} \mathcal{L}_{\text{temp}} \\
& + \lambda_{\text{con}} \mathcal{L}_{\text{con}}^{s,\text{id}}.
\end{aligned} \tag{49}$$

The first two terms are obtained directly from the variational lower bound. The commitment loss makes the lookup-free identity representation trainable, the entropy codebook-utilization loss encourages confident and balanced use of the binary identity code space, and the temporal identity-consistency loss prevents the same object's identity representation from drifting across frames. The contrastive identity–state separation loss discourages the continuous state latent from encoding identity information already captured by $\boldsymbol{z}_{\text{id}}$. We minimize this objective with respect to $(\phi_s, \phi_z, \theta, W_{\text{id}}, \boldsymbol{c}_{\text{id}})$ and the projection heads used in the contrastive separation term.

---

**Algorithm 1** Memory-Guided Identity Encoding with LFQ Tokenization

---

1: **Input:** Video $\mathcal{V} = \{x_t\}_{t=1}^T$
2: **Parameters:** LFQ temperature $\tau$, commitment weight $\beta$, entropy weight $\lambda_{\text{entropy}}$
3: **Initialize:** Class-agnostic segmenter Seg, identity encoder $q_{\phi_z}$, LFQ head $g_{\phi_z}$, identity projection $(W_{\text{id}}, c_{\text{id}})$, memory bank $\mathcal{B} \leftarrow \emptyset$

4: *// 1. Extract object proposals and proposal-level identity evidence*
5: **for** $t = 1$ to $T$ **do**
6: $\quad \mathcal{O}_t \leftarrow \text{Seg}(x_t)$ $\qquad\qquad\qquad\qquad\qquad\qquad\qquad\qquad\qquad\qquad\qquad \triangleright \mathcal{O}_t = \{o_{t,k}\}_{k=1}^{K_t}$
7: $\quad$ **for** each proposal $o_{t,k} = (b_{t,k}, m_{t,k}, v_{t,k}) \in \mathcal{O}_t$ **do**
8: $\quad\quad x_{t,k}^\alpha \leftarrow \text{RGB-A}(x_t, m_{t,k})$ $\qquad\qquad\qquad\qquad\qquad\qquad\qquad \triangleright$ object-focused input
9: $\quad\quad z_{t,k} \leftarrow q_{\phi_z}(x_{t,k}^\alpha)$ $\qquad\qquad\qquad\qquad\qquad\qquad\qquad \triangleright$ continuous identity evidence
10: $\quad$ **end for**
11: **end for**

12: *// 2. Dynamic memory assignment of proposals to persistent object indices*
13: **for** $t = 1$ to $T$ **do**
14: $\quad$ **for** each proposal $o_{t,k}$ and each active memory $M_i \in \mathcal{B}$ **do**
15: $\quad\quad C_{k,i}^{(t)} \leftarrow \eta_{\text{app}}\left(1 - \cos(z_{t,k}, \mu_i^z)\right) + \eta_{\text{box}}\left(1 - \text{IoU}(b_{t,k}, b_i^{\text{last}})\right) + \eta_{\text{mask}}\left(1 - \text{MaskIoU}(m_{t,k}, m_i^{\text{last}})\right)$
16: $\quad$ **end for**
17: $\quad \mathcal{A}_t \leftarrow \text{HungarianAssign}(C^{(t)}, \rho)$ $\qquad\qquad\qquad\qquad\qquad \triangleright$ accept matches below threshold $\rho$
18: $\quad$ **for** each matched pair $(k, i) \in \mathcal{A}_t$ **do**
19: $\quad\quad$ Assign proposal $o_{t,k}$ to memory slot $M_i$
20: $\quad\quad$ Append $x_{t,k}^\alpha$ to video tube $\mathcal{V}^{(i)}$
21: $\quad\quad$ Store indexed proposal $o_{t,i} \leftarrow o_{t,k}$
22: $\quad\quad$ Update memory summary $\mu_i^z$, $b_i^{\text{last}}$, and $m_i^{\text{last}}$
23: $\quad$ **end for**
24: $\quad$ **for** each unmatched proposal $o_{t,k}$ **do**
25: $\quad\quad$ Create new memory slot $M_j$
26: $\quad\quad \mathcal{V}^{(j)} \leftarrow \{x_{t,k}^\alpha\}$
27: $\quad\quad$ Store indexed proposal $o_{t,j} \leftarrow o_{t,k}$
28: $\quad\quad$ Initialize $\mu_j^z \leftarrow z_{t,k}$, $b_j^{\text{last}} \leftarrow b_{t,k}$, $m_j^{\text{last}} \leftarrow m_{t,k}$
29: $\quad\quad$ Insert $M_j$ into $\mathcal{B}$
30: $\quad$ **end for**
31: **end for**

32: *// 3. Track-wise LFQ identity tokenization*
33: $\mathcal{L}_{\text{commit}} \leftarrow 0$
34: **for** each memory slot $M_i \in \mathcal{B}$ **do**
35: $\quad \bar{z}^{(i)} \leftarrow \frac{1}{|\mathcal{V}^{(i)}|} \sum_{x_{t,k}^\alpha \in \mathcal{V}^{(i)}} q_{\phi_z}(x_{t,k}^\alpha)$ $\qquad\qquad\qquad\qquad \triangleright$ aggregate object evidence
36: $\quad a^{(i)} \leftarrow g_{\phi_z}(\bar{z}^{(i)})$ $\qquad\qquad\qquad\qquad\qquad\qquad\qquad\qquad \triangleright a^{(i)} \in \mathbb{R}^B$
37: $\quad b^{(i)} \leftarrow 2\mathbf{1}[a^{(i)} \geq 0] - 1$ $\qquad\qquad\qquad\qquad\qquad\qquad\qquad \triangleright$ binary LFQ code
38: $\quad \tilde{b}^{(i)} \leftarrow a^{(i)} + \text{sg}\left[b^{(i)} - a^{(i)}\right]$ $\qquad\qquad\qquad\qquad\qquad \triangleright$ straight-through estimator
39: $\quad z_{\text{id}}^{(i)} \leftarrow W_{\text{id}}\tilde{b}^{(i)} + c_{\text{id}}$
40: $\quad \mathcal{L}_{\text{commit}} \leftarrow \mathcal{L}_{\text{commit}} + \left\|a^{(i)} - \text{sg}[b^{(i)}]\right\|_2^2$
41: **end for**
42: $\mathcal{L}_{\text{commit}} \leftarrow |\mathcal{B}|^{-1} \mathcal{L}_{\text{commit}}$

43: *// 4. LFQ entropy code-utilization regularization*
44: **for** each memory slot $M_i \in \mathcal{B}$ **do**
45: $\quad p^{(i)} \leftarrow \sigma(a^{(i)}/\tau)$
46: **end for**
47: $\bar{p} \leftarrow |\mathcal{B}|^{-1} \sum_i p^{(i)}$
48: $\mathcal{L}_{\text{entropy}} \leftarrow |\mathcal{B}|^{-1} \sum_i \sum_{r=1}^B h(p_r^{(i)}) - \sum_{r=1}^B h(\bar{p}_r)$

49: **Output:** Memory-indexed proposals $\{o_{t,i}\}$, video tubes $\{\mathcal{V}^{(i)}\}$, identity codes $\{z_{\text{id}}^{(i)}\}$, $\mathcal{L}_{\text{commit}}$, $\mathcal{L}_{\text{entropy}}$

---

# B    Appendix: Algorithms

**Algorithm 2** Proposal-Conditioned State Encoding, Memory-Paired Decoding, and Optimization

1: **Input:** Video $\mathcal{V} = \{x_t\}_{t=1}^T$, memory-indexed proposals $\{o_{t,i}\}$, identity codes $\{z_{\mathrm{id}}^{(i)}\}$
2: **Parameters:** $\lambda_s$, $\beta$, $\lambda_{\mathrm{entropy}}$, $\lambda_{\mathrm{con}}$, margin $m$
3: **Initialize:** Frozen visual backbone $F_\psi$, geometry encoder $\gamma$, state encoder $E_{\mathrm{st}}$, decoder $D_\theta$, projection heads $r_s$ and $r_{\mathrm{id}}$
4: **while** not converged **do**
5:  $\quad$ // 1. Sample training frames
6:  $\quad$ Sample sub-sequence $\tau_s \subset \{1, \ldots, T\}$
7:  $\quad$ $\mathcal{L}_{\mathrm{rec}} \leftarrow 0$, $\mathcal{L}_{\mathrm{KL}} \leftarrow 0$, $\mathcal{L}_{\mathrm{con}}^{s,\mathrm{id}} \leftarrow 0$

8:  $\quad$ // 2. Proposal-conditioned state encoding
9:  $\quad$ **for** $t \in \tau_s$ **do**
10: $\quad\quad$ $H_t \leftarrow F_\psi(x_t)$ $\qquad\qquad\qquad\qquad\qquad\qquad\qquad\qquad\qquad\qquad$ ▷ dense ViT/DINOv2 feature map
11: $\quad\quad$ **for** each memory slot $i$ visible in frame $t$ **do**
12: $\quad\quad\quad$ $o_{t,i} = (b_{t,i}, m_{t,i}, v_{t,i})$
13: $\quad\quad\quad$ $G_{t,i} \leftarrow \mathrm{RoIAlign}(H_t, b_{t,i})$ $\qquad\qquad\qquad\qquad\qquad\qquad$ ▷ $G_{t,i} \in \mathbb{R}^{R \times R \times D}$
14: $\quad\quad\quad$ **if** mask $m_{t,i}$ is available **then**
15: $\quad\quad\quad\quad$ $\tilde{m}_{t,i} \leftarrow \mathrm{Resize}(m_{t,i}, R, R)$
16: $\quad\quad\quad$ **else**
17: $\quad\quad\quad\quad$ $\tilde{m}_{t,i} \leftarrow \mathbf{1}_{R \times R}$
18: $\quad\quad\quad$ **end if**
19: $\quad\quad\quad$ $\bar{G}_{t,i} \leftarrow \tilde{m}_{t,i} \odot G_{t,i}$ $\qquad\qquad\qquad\qquad\qquad\qquad\qquad\quad$ ▷ soft proposal gate
20: $\quad\quad\quad$ $g_{t,i} \leftarrow \gamma(b_{t,i}, m_{t,i})$
21: $\quad\quad\quad$ $(\mu_{t,i}, \log \sigma_{t,i}^2) \leftarrow E_{\mathrm{st}}\left(\mathrm{Pool}(\bar{G}_{t,i}), g_{t,i}\right)$
22: $\quad\quad\quad$ $\epsilon \sim \mathcal{N}(0, I)$
23: $\quad\quad\quad$ $s^{(t,i)} \leftarrow \mu_{t,i} + \sigma_{t,i} \odot \epsilon$
24: $\quad\quad\quad$ $\mathcal{L}_{\mathrm{KL}} \leftarrow \mathcal{L}_{\mathrm{KL}} + D_{\mathrm{KL}}\left(q_{\phi_s}(s^{(t,i)} \mid x_t, o_{t,i}) \,\|\, p(s)\right)$

25: $\quad\quad\quad$ // 3. Memory-paired identity–state decoding
26: $\quad\quad\quad$ $u^{(t,i)} \leftarrow [z_{\mathrm{id}}^{(i)}; s^{(t,i)}]$ $\qquad\qquad\qquad\qquad\qquad\qquad$ ▷ paired only by memory index $i$
27: $\quad\quad\quad$ $(\hat{v}^{(t,i)}, \hat{\alpha}^{(t,i)}) \leftarrow D_\theta(u^{(t,i)})$

28: $\quad\quad\quad$ // 4. Identity–state separation loss
29: $\quad\quad\quad$ $\mathbf{u}_{t,i} \leftarrow \frac{r_s(s^{(t,i)})}{\|r_s(s^{(t,i)})\|_2}$
30: $\quad\quad\quad$ $\mathbf{v}_i \leftarrow \frac{r_{\mathrm{id}}(z_{\mathrm{id}}^{(i)})}{\|r_{\mathrm{id}}(z_{\mathrm{id}}^{(i)})\|_2}$
31: $\quad\quad\quad$ $\mathcal{L}_{\mathrm{con}}^{s,\mathrm{id}} \leftarrow \mathcal{L}_{\mathrm{con}}^{s,\mathrm{id}} + \left[\max\left(0, \mathbf{u}_{t,i}^\top \mathrm{sg}[\mathbf{v}_i] - m\right)\right]^2$
32: $\quad\quad$ **end for**
33: $\quad\quad$ $\hat{x}_t \leftarrow \sum_i \mathrm{softmax}_i\left(\hat{\alpha}^{(t,i)}\right) \hat{v}^{(t,i)}$
34: $\quad\quad$ $\mathcal{L}_{\mathrm{rec}} \leftarrow \mathcal{L}_{\mathrm{rec}} + \|x_t - \hat{x}_t\|_2^2$
35: $\quad$ **end for**

36: $\quad$ // 5. Normalize losses
37: $\quad$ $\mathcal{L}_{\mathrm{KL}} \leftarrow N_{\mathrm{vis}}^{-1} \mathcal{L}_{\mathrm{KL}}$
38: $\quad$ $\mathcal{L}_{\mathrm{con}}^{s,\mathrm{id}} \leftarrow N_{\mathrm{vis}}^{-1} \mathcal{L}_{\mathrm{con}}^{s,\mathrm{id}}$

39: $\quad$ // 6. Full optimization objective
40: $\quad$ $\mathcal{L} \leftarrow \mathcal{L}_{\mathrm{rec}} + \lambda_s \mathcal{L}_{\mathrm{KL}} + \beta \mathcal{L}_{\mathrm{commit}} + \lambda_{\mathrm{entropy}} \mathcal{L}_{\mathrm{entropy}} + \lambda_{\mathrm{con}} \mathcal{L}_{\mathrm{con}}^{s,\mathrm{id}}$
41: $\quad$ Backpropagate $\nabla \mathcal{L}$ and update trainable parameters
42: **end while**

43: **Output:** Trained identity encoder, state encoder, LFQ projection, memory-indexed identity codes, and decoder

# C Datasets

This appendix summarizes the datasets used in the experiments. The datasets are organized according to the four evaluation settings in Sec. 4: object identity stability, identity–state factorization, out-of-distribution grounding and sample-efficient transfer, and visual reasoning transfer.

## C.1 Datasets for Object Identity Stability

### C.1.1 MS COCO

MS COCO Lin et al. (2014) is used to evaluate category-level representation stability on real-world images. Since COCO contains static images and does not provide temporal object tracks, we do not use it to evaluate physical identity persistence over time. Instead, object instances are grouped by semantic category. For each image, ground-truth instance masks are matched to predicted slots using mask IoU, and the corresponding slot embeddings are used to measure category-level representation stability.

### C.1.2 MOVi-C

MOVi-C Greff et al. (2022) is used to evaluate temporal object identity stability in synthetic multi-object videos. It contains scenes with multiple moving objects and provides frame-level instance masks, visibility annotations, and persistent object identifiers. These annotations allow us to evaluate whether the same physical object remains assigned to a consistent slot and maintains a stable latent representation across frames.

### C.1.3 MOVi-E

MOVi-E Greff et al. (2022) is used as a more challenging temporal identity benchmark. Compared with MOVi-C, MOVi-E contains denser scenes with more objects per video. The dataset provides persistent object IDs, frame-level masks, and visibility information, enabling evaluation of slot assignment consistency, representation deviation, and identity retrieval across time.

## C.2 Datasets for Identity–State Factorization

### C.2.1 MOVi-A

MOVi-A is used to evaluate identity–state factorization under controlled synthetic conditions. It provides object-level information that allows us to measure whether the identity token remains stable for the same physical object while the state representation varies with frame-dependent changes.

### C.2.2 MOVi-B

MOVi-B extends the controlled setting of MOVi-A with additional object and motion variability. We use MOVi-A/B to test whether $\mathbf{z}_{\mathrm{id}}$ captures persistent object identity and whether $s_t$ captures time-varying state factors such as position, motion, orientation, visibility, and appearance changes.

## C.3 Dataset for Out-of-distribution Grounding and Transfer

### C.3.1 OVIS

OVIS Qi et al. (2022a) is used for zero-shot out-of-distribution grounding and sample-efficient transfer. It is a real-world occluded video instance segmentation benchmark containing camera motion, clutter, object occlusion, and object disappearance/reappearance. OVIS provides temporally consistent instance annotations, which allow us to evaluate object grouping, mask grounding, identity-token stability, and identity retrieval across visible annotated frames.

For the few-shot transfer setting, we freeze the learned representation and train lightweight target heads using a small number of labelled OVIS tracks, with $k \in \{10, 50, 100\}$.

### C.4 Visual Reasoning and Transfer Benchmark

### C.4.1 FloatingMNIST-2

FloatingMNIST-2 (FMNIST2) is used to evaluate downstream visual reasoning and few-shot rule transfer. Each sample contains two MNIST digits placed on a $64 \times 64$ canvas. We use two task variants:

- **FMNIST2-Add**: the target is the sum of the two digits;

- **FMNIST2-Sub**: the target is the absolute difference between the two digits.

Models are trained on either FMNIST2-Add or FMNIST2-Sub and evaluated on both in-domain performance and few-shot cross-task transfer with $k=100$ labelled target examples.

### C.5 Evaluation Protocol and Preprocessing

For video datasets, we sample fixed-length clips and resize frames to the input resolution used by the model. Pixel intensities are normalized to $[0, 1]$. Unless otherwise specified, ground-truth instance masks and object IDs are used only for evaluation and not as direct supervision during representation learning.

For MS COCO, evaluation is performed at the instance level and then aggregated by semantic category. For MOVi-C and MOVi-E, identity stability is evaluated within each video using persistent object IDs and visible frame-level masks. For MOVi-A/B, identity–state factorization is evaluated by measuring the stability of $\mathbf{z}_{\mathrm{id}}$ and the temporal variation of $s_t$. For OVIS, predicted objects are matched to annotated masks using mask IoU, and identity persistence is measured across visible annotated frames.

Additional implementation details and metric definitions are provided in Appendix D.

# D Evaluation Metrics

This appendix defines the evaluation metrics used in Sec. 4. The metrics are organized according to the four experimental settings: object identity stability, identity–state factorization, out-of-distribution grounding and sample-efficient transfer, and visual reasoning transfer.

For video datasets, identity-related metrics are computed within each video using the dataset-provided object annotations. MOVi and OVIS object identifiers are persistent within a video but are not shared across videos; therefore, these metrics evaluate within-video physical object identity rather than cross-video semantic recognition. For MS COCO, which contains static images and does not provide temporal object tracks, identity stability is evaluated at the category level.

## D.1 Object–Slot Matching

For each visible object $O_i$ at frame $t$, we assign the object to the predicted slot with maximum mask overlap:

$$s_t(i) = \arg\max_k \text{IoU}(M_i^t, A_k^t),$$

where $M_i^t$ is the ground-truth object mask and $A_k^t$ is the predicted slot mask or attention map. The set of visible frames for object $O_i$ is denoted by $\mathcal{T}_i$.

For MS COCO, the same matching rule is applied to static images by matching each ground-truth instance mask to the predicted slot with the highest mask IoU. Since COCO does not contain temporal tracks, the resulting slot embeddings are used only for category-level representation stability.

For OVIS, predicted objects are matched to ground-truth instance masks using Hungarian matching with mask IoU. A match is accepted only when the IoU exceeds the threshold $\tau_{\text{IoU}}$.

## D.2 Slot Assignment Consistency

Slot Assignment Consistency (SAC) measures whether the same physical object is assigned to the same slot throughout its visible trajectory. For each object $O_i$, we first compute its dominant slot:

$$s_i^\star = \text{mode}(\{s_t(i)\}_{t \in \mathcal{T}_i}).$$

SAC is then defined as:

$$\text{SAC}_i = \frac{1}{|\mathcal{T}_i|} \sum_{t \in \mathcal{T}_i} \mathbb{1}[s_t(i) = s_i^\star].$$

Higher SAC indicates stronger object-slot persistence. SAC is reported for the MOVi temporal identity stability experiment.

## D.3 Representation Deviation

Representation Deviation (RD) measures how much the representation associated with the same object changes across observations. Let $z_i^t$ be the representation of object $O_i$ at frame $t$, obtained from its matched slot. The mean representation is:

$$\bar{z}_i = \frac{1}{|\mathcal{T}_i|} \sum_{t \in \mathcal{T}_i} z_i^t.$$

RD is then:

$$\text{RD}_i = \frac{1}{|\mathcal{T}_i|} \sum_{t \in \mathcal{T}_i} \|z_i^t - \bar{z}_i\|_2.$$

Lower RD indicates a more stable object representation.

For MOVi-C and MOVi-E, RD measures temporal representation stability for the same physical object. For MS COCO, RD is computed over matched object representations grouped by semantic category, since temporal identity tracks are unavailable.

### D.4 Identification Rate

Identification Rate at rank 1 (IDR@1) evaluates whether an object representation retrieves the correct identity under nearest-neighbour matching.

For temporal video datasets, given a query representation $z_i^t$, the retrieved object at a later visible frame $t + \Delta$ is:

$$\hat{i} = \arg \max_j \cos(z_i^t, z_j^{t+\Delta}).$$

The retrieval is correct if $\hat{i} = i$:

$$\text{IDR@1} = \frac{1}{|\mathcal{Q}|} \sum_{(i,t,\Delta) \in \mathcal{Q}} \not\Vdash \left[ \arg \max_j \cos(z_i^t, z_j^{t+\Delta}) = i \right].$$

Here, $\mathcal{Q}$ is the set of valid query pairs where the object is visible at both frames. Higher IDR@1 indicates stronger identity preservation.

For MS COCO, IDR@1 is computed at the category level: a retrieved representation is counted as correct when it belongs to the same semantic category as the query representation.

### D.5 Identity–State Factorization Metrics

Identity Deviation (IDev) measures how much the identity token changes for the same physical object over time:

$$\text{IDev}_i = \frac{1}{|\mathcal{T}_i|} \sum_{t \in \mathcal{T}_i} \left\| z_{\text{id}}^t(i) - \bar{z}_{\text{id}}(i) \right\|_2,$$

where

$$\bar{z}_{\text{id}}(i) = \frac{1}{|\mathcal{T}_i|} \sum_{t \in \mathcal{T}_i} z_{\text{id}}^t(i).$$

Lower IDev means that the identity token remains more stable across the video.

State Variation (SVar) measures whether the state representation changes over time:

$$\bar{s}(i) = \frac{1}{|\mathcal{T}_i|} \sum_{t \in \mathcal{T}_i} s_t(i),$$

$$\text{SVar}_i = \frac{1}{|\mathcal{T}_i|} \sum_{t \in \mathcal{T}_i} \left\| s_t(i) - \bar{s}(i) \right\|_2.$$

Higher SVar indicates that the state branch captures temporal changes such as position, motion, orientation, visibility, and appearance variation.

The identity–state factorization experiment reports IDev and SVar on MOVi-A/B.

### D.6 Out-of-distribution Grounding and Transfer Metrics

OOD FG-ARI measures foreground object grouping quality on OVIS. It evaluates how well predicted object groupings align with annotated foreground object instances, ignoring background pixels. Higher OOD FG-ARI indicates better object grouping under distribution shift.

OOD mIoU measures mask grounding quality on OVIS. It is computed between matched predicted masks and ground-truth instance masks. Higher OOD mIoU indicates better spatial grounding.

OOD IDR@1 applies the identification-rate metric to OVIS. An identity token from frame $t$ must retrieve the same object at a later visible annotated frame $t'$. Higher OOD IDR@1 indicates stronger identity persistence under real-world appearance, clutter, camera motion, and occlusion shifts.

OOD IDev applies the identity-deviation metric to OVIS:

$$\text{IDev}_i = \frac{1}{|\mathcal{T}_i|} \sum_{t \in \mathcal{T}_i} \left\| z_{\text{id}}^t(i) - \bar{z}_{\text{id}}(i) \right\|_2.$$

Lower OOD IDev means that the same object keeps a more stable identity representation across visible annotated frames.

The source-to-target generalization gap is reported as the FG-ARI drop from the source setting to OVIS:

$$\text{Gap} = \text{FG-ARI}_{\text{source}} - \text{FG-ARI}_{\text{OVIS}}.$$

Lower gap indicates better out-of-distribution transfer.

For sample-efficient transfer, the learned representation is frozen and a lightweight target head is trained using $k \in \{10, 50, 100\}$ labelled OVIS tracks. The 10-shot score reports target accuracy with $k = 10$, and Few-shot AUC summarizes target accuracy across the tested values of $k$.

### D.7 Visual Reasoning Transfer Metrics

For FMNIST2 addition and subtraction, accuracy measures whether the model predicts the correct arithmetic output.

HMC measures rationale misalignment between the model-selected evidence and the ground-truth relevant objects. Lower HMC indicates better rationale alignment.

Rationale F1 measures the overlap between the predicted relevant objects and the ground-truth relevant objects. Higher F1 indicates more faithful object-level reasoning.

In the FMNIST2 transfer experiments, source accuracy and HMC are reported for the in-domain task, while target accuracy and rationale F1 are reported for few-shot Add→Sub and Sub→Add transfer.

# E   Training and Computational Details

We train all models using the Adam optimizer with a learning rate of $4 \times 10^{-4}$ and a batch size of 16. Unless otherwise noted, training uses early stopping with a patience of 5 and runs for at most $\min(40 \text{ epochs}, 40{,}000 \text{ steps})$. We apply a linear learning-rate warmup for the first 10,000 steps, followed by a Reduce-on-Plateau scheduler with decay factor 0.5. This configuration is used for the main PSC model as well as the comparison models, unless the original implementation of a baseline requires a different scheduler for stable reproduction.

For PSC, training is performed on fixed-length video clips, since the identity branch aggregates object-level representations across time. During training, input frames are resized to the task-specific resolution described in Appendix C, normalized to $[0, 1]$, and converted into object-focused inputs using the mask-generation pipeline described in Sec. 3. Unless otherwise noted, we use the same optimization hyperparameters for the object-discovery experiments and the downstream FMNIST2 transfer experiments, with task-specific heads trained on top of the learned object representations.

We run all experiments on a compute cluster equipped with NVIDIA Tesla T4 16GB GPUs and Intel(R) Xeon(R) Gold 6230 CPUs. Throughput is reported in iterations per second (it/s). These measurements may vary slightly across systems depending on I/O load, background processes, and implementation details, so they should be interpreted as approximate efficiency indicators rather than absolute hardware-independent benchmarks.

To support reproducibility, we will release the experimental configuration files, training scripts, inference scripts, and dataset-generation scripts used in our experiments.

**Implementation notes.**   For fair comparison, we use the official implementations of the baselines whenever available and preserve their default architectural components unless changes are required to match input resolution or evaluation protocol. When a baseline is adapted to our setting, we report this explicitly in the corresponding experiment section. For throughput reporting, all models are evaluated under the same batch size and hardware configuration.

