# OpenReview forum: "Object-Centric Representation Learning via Probabilistic Superpixel Coding"
_TMLR — Withdrawn by Authors_

### Review · Reviewer_9PjM · 2026-04-07

**Summary Of Contributions:**

This paper addresses the identity instability issue in slot‑based object‑centric representation learning by proposing Probabilistic Superpixel Coding (PSC), which replaces interchangeable slots with a shared discrete identity codebook to separate object identity from time‑varying state. The model explicitly enforces temporal identity consistency through cross‑frame matching and aggregation, while optimizing a variational objective with vector quantization and orthogonality regularization to disentangle stable identity and transient attributes. Extensive experiments on object discovery, video identity stability, representation grounding, and visual reasoning show that PSC outperforms slot‑based baselines and yields more robust, generalizable object representations.

**Audience:**

Yes

**Audience Explanation:**

The findings offer novel insights into object-centric representation learning.

**Broader Impact Concerns:**

This work focuses on fundamental object-centric representation learning for computer vision, with no direct ethical risks, harmful applications, or privacy concerns identified.

**Claims And Evidence:**

Yes

**Claims Explanation:**

The claims are thoroughly validated by rigorous experiments, comprehensive ablations, and clear comparisons with state-of-the-art baselines.

**Requested Changes:**

1. The proposed Probabilistic Superpixel Coding (PSC) relies on a fixed-size discrete codebook, which may limit its ability to generalize to open-world scenarios with unseen object categories or novel visual concepts.
2. The temporal consistency enforcement relies on cross-frame matching heuristics, which could fail in dynamic scenes with rapid object motion, occlusion, or appearance changes.
3. The experimental evaluation is primarily conducted on synthetic and controlled video datasets, with limited validation on real-world, high-resolution, and unconstrained video benchmarks.
4. The computational complexity of vector quantization and cross-frame aggregation increases significantly with longer video sequences, hindering deployment in real-time applications.
5. The ablation studies do not fully isolate the impact of individual components (e.g., orthogonality regularization vs. cross-frame matching) on the final performance, leaving unclear the marginal contribution of each design choice.
6. The generalization ability to out-of-distribution datasets (e.g., cross-domain, cross-resolution, or cross-camera settings) is not systematically evaluated, leaving the model’s robustness unvalidated.

---

> ### Author Response · Authors · 2026-05-09
> **Official Comment by Authors**
>
> We thank the reviewer for their detailed comments and constructive feedback. We very much appreciate the fact that our framework was received as novel insights into object-centric representation learning and insightful for the community.
>
> >The proposed Probabilistic Superpixel Coding (PSC) relies on a fixed-size discrete codebook, which may limit its ability to generalize to open-world scenarios with unseen object categories or novel visual concepts.
>
> We thank the reviewer for this important observation. The reviewer is correct that the original version of PSC used a fixed-size discrete codebook, which could limit open-world generalization. To address this concern, in the revised manuscript we replace the fixed lookup codebook with a lookup-free identity tokenizer. This makes the identity representation less tied to a predefined dictionary size. We also clarify that fully adaptive open-world identity expansion remains an important direction for future work.
>
> >The temporal consistency enforcement relies on cross-frame matching heuristics, which could fail in dynamic scenes with rapid object motion, occlusion, or appearance changes.
>
> We share the reviewer’s concern that simple cross-frame matching may fail under rapid motion, occlusion, or appearance change. To address this, we revised PSC to use a dynamic memory bank instead of direct cross-frame matching. The memory bank aggregates identity evidence over time and allows objects to disappear and reappear more robustly.
>
> > The experimental evaluation is primarily conducted on synthetic and controlled video datasets, with limited validation on real-world, high-resolution, and unconstrained video benchmarks.
>
> We thank the reviewer for the suggestion. We agree that evaluating on real-world video benchmarks such as DAVIS-2017 would be valuable. At this stage, we believe this is best treated as future work because such evaluation introduces additional factors such as segmentation quality, camera motion, and high-resolution computational cost. We have clarified this limitation in the revised manuscript.
>
> >The computational complexity of vector quantization and cross-frame aggregation increases significantly with longer video sequences, hindering deployment in real-time applications.
>
> We agree that scalability to long videos is important. The revised memory-bank formulation avoids exhaustive all-frame cross-frame aggregation and updates a compact set of object memories online. Still, PSC is not currently intended as a real-time long-video system
>
> >The ablation studies do not fully isolate the impact of individual components (e.g., orthogonality regularization vs. cross-frame matching) on the final performance, leaving unclear the marginal contribution of each design choice.
>
> We agree that ablations provide important insight into the contribution of each component. We have added ablations on the memory bank and codebook size, and clarified the role of the commitment loss, entropy regularization, temporal consistency loss, and state KL term. These additions better isolate the effect of PSC’s main design choices.
>
> >The generalization ability to out-of-distribution datasets (e.g., cross-domain, cross-resolution, or cross-camera settings) is not systematically evaluated, leaving the model’s robustness unvalidated.
>
> We thank the reviewer for raising this point. We agree that systematic OOD evaluation across domains, resolutions, and camera settings would strengthen the analysis. The revised manuscript now avoids overclaiming OOD robustness and discusses broader cross-domain evaluation as future work.

---

> > ### Comment · Action_Editor_qcwc · 2026-05-13
> >
> > Dear Reviewer,
> >
> > The authors have submitted their rebuttal. Please proceed with the discussion and finalize your recommendations. If any additional clarification is required, feel free to raise your questions to the authors.
> >
> > Thank you for your contributions to TMLR.
> >
> > Best, AE

---

> ### Author Response · Authors · 2026-05-13
> **Official Comment by Authors**
>
> Dear Reviewer, We have submitted responses to the reviewer comments, but I have not yet uploaded a revised manuscript. The requested revisions involve substantial changes and require additional time to complete carefully. We are currently working on the updated version and plan to upload the revised manuscript as soon as the revisions are finalized. In the meantime, We wanted to respond promptly during the discussion period and clarify how I intend to address the reviewers’ concerns.

---

> ### Author Response · Authors · 2026-05-21
> **Official Comment by Authors**
>
> Dear Reviewers,
>
> We have uploaded a revised version of the manuscript addressing the previous review comments.
>
> Best,
> The Authors

---

### Review · Reviewer_tQU1 · 2026-05-07

**Summary Of Contributions:**

This paper proposes a model called Probabilistic Superpixel Coding (PSC) for consistently binding object representations over time. The model factorizes each object representation into an identity vector and a state vector. Experimental results demonstrate that the proposed method can benefit multiple downstream tasks.

**Audience:**

No

**Audience Explanation:**

I am actually not sure whether this work would attract audience interest, since I believe Slot Attention may already be capable of achieving similar object binding behavior. Please see my comments above for more details. I am open to changing my opinion if the authors can provide stronger evidence and more convincing support for their claims.

**Broader Impact Concerns:**

No concerns on the ethical implications of the work.

**Claims And Evidence:**

No

**Claims Explanation:**

1. My main concern is regarding the claim that Slot Attention cannot consistently specialize slots to individual objects. It seems possible that, if slot parameters are made fully learnable during training and fixed during inference, each slot could potentially bind consistently to a particular object.
2. In Tables 1 and 2, the performance improvement introduced by PSC appears relatively small. Are these improvements statistically significant? For example, it may be helpful to report two-tailed t-test p-values.

**Requested Changes:**

1. In the identity encoder, is it sufficiently robust to match objects across frames solely based on the first frame? If the first frame contains low-quality observations or poses, could this negatively affect the tracking consistency in subsequent frames?
2. In the identity encoder, the authors use cosine similarity for object tracking across different frames. Does this implicitly assume that the appearance features of an object remain relatively stable over time? Similarly, does the same assumption justify the temporal aggregation of embeddings along each track?
3. In the state encoder, it is unclear how the soft patch-to-object assignment matrix $M_t$ is computed in practice. More implementation details would be helpful.
4. Do the authors use a standard ViT backbone in the state encoder, or a self-supervised variant such as ViT-DINO to obtain more semantically meaningful patch representations?
5. The model first relies on an external class-agnostic segmentation model to generate binary masks. Why not directly use the features extracted by the segmentation model itself as identity features?
6. Could the authors further clarify the motivation for introducing vector quantization in this task?
7. What does “Temporal Stabilization” refer to in Table 6? In addition, Table 6 is only briefly introduced, and its results do not appear to be sufficiently discussed in the main text.

---

> ### Author Response · Authors · 2026-05-09
> **Official Comment by Authors**
>
> We thank the reviewer for their detailed comments and constructive feedback.
>
> > My main concern is regarding the claim that Slot Attention cannot consistently specialize slots to individual objects.
>
> We thank the reviewer for raising this point. We agree that learnable slot initializations can partially reduce slot symmetry in some settings. However, standard Slot Attention remains permutation-symmetric in its update rule and does not explicitly enforce that the same physical object keeps the same identity across frames. We revised the wording to avoid overclaiming and now clarify that PSC explicitly models identity persistence through a separate identity token and state vector.
>
> > In Tables 1 and 2, the performance improvement introduced by PSC appears relatively small. Are these improvements statistically significant?
>
> We agree that the static object-discovery improvements are modest. We revised the narrative to de-emphasize static object-discovery tables as the main evidence. The core claim of PSC is temporal identity stability, so the revised manuscript focuses more strongly on video-centric identity metrics and downstream compositional reasoning. We also report mean/std over multiple seeds where applicable.
>
> > In the identity encoder, is it sufficiently robust to match objects across frames solely based on the first frame?
>
> We agree that relying only on the first frame would be brittle. In the revised method, PSC no longer uses first-frame-only matching. We replace this with a dynamic memory bank that stores object-specific identity evidence over time and updates identities as objects move, disappear, or reappear.
>
> > In the identity encoder, the authors use cosine similarity for object tracking across different frames. Does this implicitly assume that the appearance features of an object remain relatively stable over time?
>
> We agree that pure cosine matching can be sensitive to large appearance changes. The revised PSC uses memory-based identity aggregation rather than only pairwise cosine matching. The identity branch captures stable object evidence, while the state branch captures transient changes such as pose, motion, and local appearance. We clarified this design choice and its limitation in the manuscript.
>
> > In the state encoder, it is unclear how the soft patch-to-object assignment matrix is computed in practice.
>
> We thank the reviewer for pointing this out. We added implementation details explaining that ViT patch features are used to construct a patch-affinity graph, followed by normalized Laplacian spectral decomposition. The resulting spectral embedding is converted into soft patch-to-object assignments and used to pool patch features into object-level state vectors.
>
> > Do the authors use a standard ViT backbone in the state encoder, or a self-supervised variant such as ViT-DINO?
>
> We agree this should be explicit. We have added the exact backbone details in the implementation section, including whether the ViT is randomly initialized or pretrained/self-supervised for each setting.
>
> > The model first relies on an external class-agnostic segmentation model to generate binary masks. Why not directly use the features extracted by the segmentation model itself as identity features?
>
> We thank the reviewer for this useful question. We use the segmentation model only to provide object masks, while PSC learns a separate identity representation. Segmentation features are optimized for mask prediction and may not provide temporally persistent identity codes or a factorized identity/state representation. We added this clarification and mention direct use of segmentation features as an interesting future ablation.
>
> > Could the authors further clarify the motivation for introducing vector quantization in this task?
>
> We added clarification that vector quantization is used to create reusable discrete identity anchors. This reduces continuous identity drift and helps separate stable identity from time-varying state. In the revised version, we use lookup-free quantization rather than a fixed lookup dictionary.
>
> > What does “Temporal Stabilization” refer to in Table 6?
>
> We clarified that “Temporal Stabilization” refers to the memory-bank identity aggregation together with the temporal consistency loss. It encourages embeddings assigned to the same object memory to remain close over time. We also expanded the discussion of Table 6 in the main text.

---

> > ### Comment · Action_Editor_qcwc · 2026-05-13
> >
> > Dear Reviewer,
> >
> > The authors have submitted their rebuttal. Please proceed with the discussion and finalize your recommendations. If any additional clarification is required, feel free to raise your questions to the authors.
> >
> > Thank you for your contributions to TMLR.
> >
> > Best, AE

---

> ### Author Response · Authors · 2026-05-21
> **Official Comment by Authors**
>
> Dear Reviewers,
>
> We have uploaded a revised version of the manuscript addressing the previous review comments.
>
> Best,
> The Authors

---

> > ### Comment · Reviewer_tQU1 · 2026-05-27
> >
> > Thanks for the rebuttal.
> > 1. My main concerns remain unresolved. As mentioned in my previous review, if the slot parameters are made fully learnable during training and fixed during inference, each slot could potentially bind consistently to a particular object. If such a simple modification to Slot Attention is already sufficient for object binding, it is unclear why a much more complex model is necessary to achieve the same goal.
> > 2. Could the authors explain in greater detail how the Perception Encoder operates? In particular, how does it extract continuous identity features?
> > 3. I believe an additional ablation study is necessary. Specifically, if the PSC directly uses the masking features from the segmentation model as identity features, what performance would be achieved? At present, there appears to be some redundancy in the model design, and the explanation provided in the rebuttal does not sufficiently address this concern.

---

> > > ### Author Response · Authors · 2026-05-29
> > > **Official Comment by Authors**
> > >
> > > Thank you for the comment. We would like to clarify that our concern is not that Slot Attention lacks learnable parameters. Slot Attention is learnable during training and fixed during inference, as is standard for neural-network models. In the original Slot Attention formulation, the slot initialization distribution has shared learnable parameters, and the attention projections, recurrent updates, and MLP updates are also learned.
> > >
> > > The issue we address is instead that slot representations are not explicitly grounded to persistent object identity. Slot Attention is designed so that slots are exchangeable: any slot can bind to any object. This is useful for object discovery, but it does not guarantee that the same physical object will keep the same representation across frames or scenes. For example, the same orange may be represented by different slots at different times, or multiple slots may represent different parts or appearances of the same orange if this improves the reconstruction objective. This can lead to multiple latent representations for the same object identity, rather than one grounded and reusable concept of that object.
> > >
> > > Therefore, simply making the slot parameters fixed at inference does not resolve the grounding problem. It only freezes the representation learned during training. If the learned slot-object correspondence is not grounded to persistent identity, then fixing it preserves the same ambiguity. Similarly, per-slot specialization may produce stable slot indices, but it does not by itself ensure that those indices correspond to grounded object identities rather than dataset-specific appearance, position, color, or reconstruction shortcuts.
> > >
> > > PSC addresses a different problem: how to map repeated observations of the same object to a stable identity representation while separating this identity from time-varying state. We do this by using grounded object proposals, temporal grouping, and LFQ identity tokenization to create a reusable identity code, while the state branch captures frame-specific variation such as pose, position, motion, occlusion, and local appearance. This follows the binding-problem perspective that object representation requires not only segregating objects, but also tying internal representations to stable entities across changing observations (Harnad, 1990; Greff et al., 2020).
> > >
> > > We agree that the Perception Encoder should be described more explicitly. In the revised manuscript, we will add implementation details explaining how object-focused RGB-A proposals are processed, which encoder features are extracted, how continuous identity evidence is produced, and how temporal aggregation and LFQ convert this evidence into a stable identity token.
> > >
> > > We also agree that the requested ablation is important. In the revised version, we will include an ablation where PSC directly uses segmentation/mask features as identity features, while keeping the memory bank, decoder, and evaluation protocol fixed. This will test whether mask features alone are sufficient, or whether the proposed Perception Encoder, temporal aggregation, and LFQ identity tokenization provide additional benefit.

---

### Review · Reviewer_n27Y · 2026-05-08

**Summary Of Contributions:**

This paper proposes an improvement in Slot Attention technique by proposing Probabilistic Superpixel Coding (PSC), which replaces the interchangeable slots with a discrete identity codebook (VQ-VAE-style) combined with a continuous state encoder. In such a design, the same object in different frames in a video is encouraged to be encoded using the same attention slot, which marginally improves performance in several video understanding and generation tasks. While the motivation of the problem is somewhat discussed, the paper does not structurally explain the limitations of previous systems which uses random slots to encode an object. Though the authors report the results in multiple different tasks and datasets, including object discovery, video-centric identity stability and robustness, evaluating representation grounding, visual reasoning and transfer, most of the qualitative improvements seem to be marginal.

**Audience:**

Yes

**Audience Explanation:**

While the topic of the paper is interesting, the current manuscript does not systematically explain the limitations of existing systems, and the quantitative improvements seem to be marginal.

**Claims And Evidence:**

No

**Claims Explanation:**

While the problem statement and general direction is interesting, the paper has limitations in multiple directions:

**Major Limitations**

(1) Firstly, the paper presents the methodological innovations in a difficult-to-understand format. The section 3.1directly throws a complex objective function in Equation 1 without providing any context of existing systems, and hence, it becomes extremely difficult to comprehend the core novelty in this equation. There is no discussion of why existing slot-based objectives fail to capture identity and how this formulation fixes that gap. The reader is expected to accept the loss function before understanding the model's purpose.

(2) In Table 1, the improvements obtained by PSC over previous systems are in the order of ~0.001 which is extremely marginal, and may not reflect in real world. Similarly, in table 2 and table 7, the core performance improvements are extremely low, questioning the efficacy of the proposed model.

(3) PSC uses external, class-agnostic segmentation masks as input. Slot-based methods (SA, DINOSAUR, CoSA, etc.) learn object discovery end-to-end without any such oracle. Evaluating PSC alongside these methods on ARI and MSE is comparing a supervised-segmentation-assisted pipeline to unsupervised learners. The comparison is fundamentally unfair and the marginal gains are largely attributable to the stronger grounding signal, not the identity codebook.

(4) The paper does not provide any qualitative results or discuss where the proposed system fails.

**Minor Limitations**
(1) The paper contains unfilled placeholders. Section 4.3 explicitly reads "[dataset names / evaluation split]". Similarly, table 6 says "Evaluation of representation grounding on [dataset names]".

Overall, the paper in its current state does not meet the evidentiary standards required for acceptance.

**Requested Changes:**

I request the authors to address the concerns raised in the major limitations section.

---

> ### Author Response · Authors · 2026-05-10
> **Official Comment by Authors**
>
> We thank the reviewer for the detailed and constructive feedback. We appreciate that the reviewer found the topic interesting, and we have revised the manuscript to better clarify the motivation, scope, and evidence for PSC.
>
> > The methodological innovations are difficult to understand, and Eq. 1 is introduced before explaining why existing slot-based objectives fail to capture identity.
>
> We agree that the method presentation needed clearer motivation. We revised Section 3 to first explain the limitation of exchangeable slot representations for persistent identity binding, then introduce PSC as a factorization of each object into a stable identity token and a time-varying state vector. The objective is now introduced after this modeling motivation, and we added explanatory text for each loss term.
>
> > The improvements in Tables 1, 2, and 7 are marginal.
>
> We agree that the static object-discovery gains are modest and should not be presented as the main contribution. We revised the narrative to de-emphasize static ARI/MSE improvements and instead focus the central evaluation on video-centric identity stability and downstream compositional reasoning, where PSC is designed to help. We also clarified that the main claim is not that PSC substantially improves all reconstruction/object-discovery metrics, but that explicit identity coding improves temporal identity consistency.
>
> > PSC uses external class-agnostic segmentation masks, making comparison with unsupervised slot-based methods unfair.
>
> We thank the reviewer for pointing this out. We revised the manuscript to make this distinction explicit. PSC should be viewed as a grounded object-centric representation method that assumes object proposals, not as a fully end-to-end unsupervised object discovery model. We now avoid presenting comparisons to SA/DINOSAUR/CoSA as purely equivalent object-discovery comparisons, and clarify that the intended comparison is about representation stability after object grounding.
>
> > The paper does not provide qualitative results or discuss failure cases.
>
> We agree. We added qualitative visualizations showing identity binding across multiple objects and added a failure-case discussion. In particular, we now discuss cases where PSC can fail due to poor segmentation masks, merged objects, missed small objects, severe occlusion, or incorrect object-count assumptions.
>
> > The paper contains unfilled placeholders such as “[dataset names / evaluation split]”.
>
> We thank the reviewer for catching this. We have removed the placeholders and replaced them with the correct dataset names, evaluation splits, and table descriptions.
>
> Overall, we revised the manuscript to make the contribution more precise: PSC is not proposed as a replacement for end-to-end unsupervised object discovery, but as a grounded object-centric representation method for improving persistent identity binding over time.

---

> > ### Comment · Action_Editor_qcwc · 2026-05-13
> >
> > Dear Reviewer,
> >
> > The authors have submitted their rebuttal. Please proceed with the discussion and finalize your recommendations. If any additional clarification is required, feel free to raise your questions to the authors.
> >
> > Thank you for your contributions to TMLR.
> >
> > Best, AE

---

> ### Author Response · Authors · 2026-05-13
> **Official Comment by Authors**
>
> Dear Reviewer, We have submitted responses to the reviewer comments, but I have not yet uploaded a revised manuscript. The requested revisions involve substantial changes and require additional time to complete carefully. We are currently working on the updated version and plan to upload the revised manuscript as soon as the revisions are finalized. In the meantime, We wanted to respond promptly during the discussion period and clarify how I intend to address the reviewers’ concerns.

---

> ### Author Response · Authors · 2026-05-21
> **Official Comment by Authors**
>
> Dear Reviewers,
>
> We have uploaded a revised version of the manuscript addressing the previous review comments.
>
> Best,
> The Authors

---

> > ### Comment · Reviewer_n27Y · 2026-06-02
> > **Response from Reviewer n27Y after the Rebuttal**
> >
> > Thanks to the authors for the rebuttal and the modifications to the manuscript. I think Section 3 (Method) is slightly better structured now. However, apart from this, the paper still has multiple critical issues:
> >
> > (1) The paper does not report any ablations on the five different objective components.
> >
> > (2) In the rebuttal, the authors mentioned, “We added qualitative visualizations showing identity binding across multiple objects.” However, the updated manuscript does not include any qualitative examples or error analysis, which is a critical issue and a false statement.
> >
> > (3) The authors mentioned four different systems for object mask computation. Which specific system was used?
> >
> > (4) The authors changed the results significantly during the rebuttal and also deleted a few tables. However, the revised manuscript does not indicate which lines are newly added, making it extremely inconvenient for the reviewer to understand what has changed during the rebuttal.
> >
> > Hence, I strongly recommend rejection of this manuscript in its current form and request that the authors incorporate all the suggestions from the reviewers’ discussions to revise the draft.

---

### Review · Reviewer_xWq7 · 2026-05-13

**Summary Of Contributions:**

The paper proposes Probabilistic Superpixel Coding (PSC), an object-centric representation learning method that replaces exchangeable slots with a discrete identity codebook and separates each object into a time-invariant identity code and a time-varying state representation. The main strength is the clear motivation around identity persistence in object-centric learning, especially for videos. The paper also evaluates object discovery, temporal identity stability, robustness to segmentation/object-count assumptions, and downstream reasoning. However, the evidence is not fully convincing: several gains over strong baselines are very small, some evaluation details appear incomplete, and the method relies heavily on external segmentation masks and fixed object-count assumptions.

**Additional Comments:**

The paper has an interesting direction and the writing is generally clear, but it currently reads like an incomplete submission in some places. The main idea could be valuable, but the empirical evidence should be made more rigorous before acceptance.

**Audience:**

Yes

**Audience Explanation:**

The topic is relevant to the TMLR audience because stable object identity, discrete object-level representations, and object-centric video learning are active research directions. The idea of replacing slot identity emergence with an explicit codebook and temporal aggregation is interesting, especially for researchers working on object-centric learning, representation grounding, and compositional reasoning. The paper also acknowledges important limitations, including dependence on external segmentation quality, object-count priors, and additional overhead.

**Broader Impact Concerns:**

I do not see major broader-impact concerns requiring rejection. However, since the method depends on external segmentation and identity tracking, the authors should briefly discuss possible failure modes in real-world visual systems, especially when segmentation errors cause incorrect object identity assignment.

**Claims And Evidence:**

No

**Claims Explanation:**

The paper provides many experiments, but the support for the main claims is currently not fully convincing. The improvements on standard object-discovery and reasoning benchmarks are often marginal, e.g., PSC only slightly improves over CoSA/AdaSlot/SPOT in several tables. More importantly, some parts of the evaluation appear under-specified or unfinished, such as the grounding evaluation using “[dataset names / evaluation split]” and Table 6 being reported on “[dataset names].” The video identity metrics are relevant to the paper’s main claim, but they need clearer protocol details, standard deviations, and stronger justification that baselines are fairly reproduced under the same proposal/mask pipeline.

**Requested Changes:**

Critical changes:

1. Complete and clarify all unfinished experimental descriptions, especially the grounding evaluation where dataset names and splits are currently placeholders.
2. Provide full experimental details for the video identity metrics, including how tracks are matched, whether ground-truth masks or predicted masks are used, and whether baselines use the same proposal pipeline.
3. Report mean ± standard deviation consistently for all main results, especially Table 3, where the caption says results should be reported over multiple seeds but the table only gives single numbers.
4. Strengthen the ablation study to isolate the contribution of the identity codebook, temporal aggregation, state/identity factorization, and external masks.
5. Clarify the novelty relative to grounded OCL methods, VQ-based latent models, and video object-centric models, since PSC combines several existing components.

Changes that would strengthen the work:

1. Add qualitative visualizations of identity consistency across frames, including failure cases under occlusion and object re-entry.
Include runtime, memory, and segmentation overhead comparisons.
2. Discuss codebook usage statistics, collapse prevention, and sensitivity to codebook size.
3. Evaluate on longer and more complex videos to better support the identity persistence claim.

---

> ### Comment · Action_Editor_qcwc · 2026-05-13
>
> Dear authors,
>
> One more review has been posted; please address them and submit your rebuttal and your revised manuscript soon.
> Best,
> AE

---

> > ### Author Response · Authors · 2026-05-13
> > **Official Comment by Authors**
> >
> > Dear Reviewer,
> > We have submitted responses to the reviewer comments, but I have not yet uploaded a revised manuscript. The requested revisions involve substantial changes and require additional time to complete carefully. We are currently working on the updated version and plan to upload the revised manuscript as soon as the revisions are finalized.
> > In the meantime, We wanted to respond promptly during the discussion period and clarify how I intend to address the reviewers’ concerns.

---

> ### Author Response · Authors · 2026-05-13
> **Official Comment by Authors**
>
> We thank the reviewer for the detailed and constructive feedback. We appreciate that the reviewer found the motivation around persistent object identity clear and the direction relevant to the TMLR audience.
>
> > Complete and clarify all unfinished experimental descriptions, especially the grounding evaluation where dataset names and splits are currently placeholders.
>
> We agree. The placeholders should not have remained in the submission. We will revise the manuscript to complete all dataset names, evaluation splits, and grounding-evaluation protocols, and ensure that all tables and captions correspond to finalized experiments.
>
> > Provide full experimental details for the video identity metrics, including how tracks are matched, whether ground-truth masks or predicted masks are used, and whether baselines use the same proposal pipeline.
>
> We agree. We will expand the experimental protocol to describe how tracks are matched, which mask source is used in each setting, and how IDF1, HOTA, identity switches, Re-ID@1, and occlusion recovery are computed. We will also clarify that baselines are evaluated under the same proposal pipeline whenever applicable.
>
> > Report mean ± standard deviation consistently for all main results, especially Table 3, where the caption says results should be reported over multiple seeds but the table only gives single numbers.
>
> We agree. In the revised manuscript, we will report mean ± standard deviation consistently for the main results, including Table 3 and the other identity, ablation, and transfer experiments.
>
> > Strengthen the ablation study to isolate the contribution of the identity codebook, temporal aggregation, state/identity factorization, and external masks.
>
> We agree. We will strengthen the ablation study to separately analyze the identity codebook, temporal aggregation, state/identity factorization, and the use of external masks. This will make clearer which components contribute to identity persistence and which contribute to object grounding.
>
> > Clarify the novelty relative to grounded OCL methods, VQ-based latent models, and video object-centric models, since PSC combines several existing components.
>
> We thank the reviewer for this suggestion. We will clarify that PSC does not claim that object proposals, vector quantization, or video OCL are individually new. The contribution is their combination in a unified object-centric framework that explicitly separates persistent discrete identity from time-varying state for temporal identity stability.
>
> > Add qualitative visualizations of identity consistency across frames, including failure cases under occlusion and object re-entry. Include runtime, memory, and segmentation overhead comparisons.
>
> We agree. We will add qualitative visualizations showing identity consistency across frames, including examples with occlusion and object re-entry. We will also include failure cases and add runtime, memory, and segmentation/proposal overhead comparisons.
>
> > Discuss codebook usage statistics, collapse prevention, and sensitivity to codebook size.
>
> We agree. We will add codebook usage statistics, discuss collapse prevention, and include sensitivity analysis for codebook size to clarify how this hyperparameter affects identity stability and representation quality.
>
> > Evaluate on longer and more complex videos to better support the identity persistence claim.
>
> We agree that longer and more complex videos would further strengthen the identity-persistence claim. We will include such experiments if feasible within the revision timeline. Otherwise, we will explicitly qualify the scope of the current results and discuss long-video evaluation as an important future-work direction.

---

> ### Author Response · Authors · 2026-05-21
> **Official Comment by Authors**
>
> Dear Reviewers,
>
> We have uploaded a revised version of the manuscript addressing the previous review comments.
>
> Best,
> The Authors

---

### Author Response · Authors · 2026-05-10
**Official Comment by Authors**

We thank the reviewers for their constructive feedback. We have posted point-by-point responses to the reviews and are currently incorporating the corresponding changes into the revised manuscript. We will upload the revised PDF shortly and will leave a follow-up comment once the revision is available for the reviewers and the Action Editor.

---

### Note · Authors · 2026-06-02

**Comment:**

The authors are withdrawing the submission in order to substantially revise the manuscript and conduct additional experiments.

**Withdrawal Confirmation:**

I have read and agree with the venue's withdrawal policy on behalf of myself and my co-authors.